# Evolutionary differentiation of androgen receptor is responsible for sexual characteristic development in a teleost fish

Yukiko Ogino [1,2] ✉, Satoshi Ansai [3,19], Eiji Watanabe [4,5], Masaki Yasugi [6,20], Yukitoshi Katayama [7], Hirotaka Sakamoto [7], Keigo Okamoto [1], Kataaki Okubo [8], Yasuhiro Yamamoto [9], Ikuyo Hara [10,11], Touko Yamazaki [10], Ai Kato [11], Yasuhiro Kamei [5,12], Kiyoshi Naruse [5,10,11], Kohei Ohta [13], Hajime Ogino [14], Tatsuya Sakamoto [7], Shinichi Miyagawa [15], Tomomi Sato [16], Gen Yamada [17], Michael E. Baker [18] & Taisen Iguchi [16]

Teleost fishes exhibit complex sexual characteristics in response to androgens, such as fin enlargement and courtship display. However, the molecular mechanisms underlying their evolutionary acquisition remain largely unknown. To address this question, we analyse medaka (*Oryzias latipes*) mutants deficient in teleost-specific androgen receptor ohnologs (*ara* and *arb*). We discovered that neither *ar* ohnolog was required for spermatogenesis, whilst they appear to be functionally redundant for the courtship display in males. However, both were required for reproductive success: *ara* for tooth enlargement and the reproductive behaviour eliciting female receptivity, *arb* for male-specific fin morphogenesis and sexual motivation. We further showed that differences between the two *ar* ohnologs in their transcription, cellular localisation of their encoded proteins, and their downstream genetic programmes could be responsible for the phenotypic diversity between the *ara* and *arb* mutants. These findings suggest that the *ar* ohnologs have diverged in two ways: first, through the loss of their roles in spermatogenesis and second, through gene duplication followed by functional differentiation that has likely resolved the pleiotropic roles derived from their ancestral gene. Thus, our results provide insights into how genome duplication impacts the massive diversification of sexual characteristics in the teleost lineage.

Vertebrates exhibit a great variety of sexually dimorphic morphological and behavioural traits that are thought to have evolved through sexual selection. Sexual dimorphism in vertebrates has been extensively studied in rodents, especially in the context of male-biased sexual characteristics, such as external genitalia, sex accessory organ development, and reproductive behaviours[1,2]. However, few empirical studies have investigated the molecular mechanisms underlying the evolutionary acquisition of diverse sexually dimorphic traits in vertebrates. Teleosts show extreme sexual dimorphism[3], such as an elongated median fin[4], copulatory organs[5,6], and nuptial coloration[7], and behavioural traits including nest building[8], courtship[9], and aggressive acts[10]. Such male traits increase male reproductive success but often harm females because of the sex-specific fitness optima of these traits. Therefore, the genetic correlation between the two sexes constrains the evolution of sexual dimorphism, which generates intra-locus sexual conflict[11,12]. Although male-biased gene

---

expression by androgens is a possible mechanism to resolve such sexual conflict by allowing the expression of morphological and behavioural traits specifically in mature male vertebrates[13–16], little is known about the relationship between the genetic programme regulated by androgens and the evolutionary diversification of sexually dimorphic traits.

Phenotypic and physiological responses to androgens are mediated by the androgen receptor (Ar), which belongs to the nuclear receptor family, a diverse group of transcription factors that originated in multicellular animals[17–20]. The Ar is composed of three major functional domains, a hypervariable N-terminal domain (NTD), a central highly conserved DNA binding domain (DBD) consisting of two zinc finger motifs, and a COOH-terminal ligand binding domain (LBD)[17–20] (Supplementary Fig. 1). In mammals, testosterone (T) and 5α-dihydrotestosterone are effective ligands for AR[21]. 11Ketotestosterone (11KT) is a potent androgen in teleosts[22]. Traditional studies using mouse models have shown that *AR* knockout (*AR* KO) results in the demasculinisation of external genitalia, agenesis of accessory sex organs, and arrest of spermatogenesis[1]. Such pleiotropic functions[23] might have constrained the molecular evolution of the *AR* gene, although androgen-mediated gene expression can solve sexual conflict.

Most teleosts have two distinct *ar* ohnologs—*ara* and *arb*—generated by teleost-specific whole-genome duplication (TSGD)[24,25]. The medaka *ara* and *arb* were mapped to chromosomes 10 and 14, respectively, with a conserved synteny relative to a single region locating the *AR* gene in human chromosome X, suggesting that the teleost *ar* gene duplication occurred as the result of TSGD[25]. TSGD is an evolutionarily recent whole-genome duplication that occurred ~350 Ma after the split of non-teleost actinopterygian lineages (namely, bichir, sturgeon, gar, and bowfin) from the teleost lineage, but before the divergence of Elopomorpha (i.e. Japanese eel) and Osteoglossomorpha (i.e. silver arowana)[26,27]. Because gene duplication results in a decrease in the negative selection pressure and can drive the establishment of lineage-specific traits with novel biological functions[28], the *ar* ohnologs could have contributed to the diversification of masculine sexual characteristics found in teleosts, owing to the reduction of the pleiotropic constraint on them.

Molecular evolutionary analysis of the teleost lineage has revealed the asymmetric evolution of *ar* ohnologs, including the accumulation of more novel substitutions in *ara* than in *arb* after the divergence of Elopomorpha, and that the lineage-specific loss of *ara* occurred independently in Otocephala such as zebrafish and Salmoniformes such as rainbow trout (Supplementary Fig. 1)[24,25,29–31]. Our in vitro analysis of the protein function of medaka Ars indicates that Arb has properties similar to those of other vertebrate Ars, but Ara acquired a new function as a hyperactive form of Ar, showing higher ligand-dependent transactivation capacity and constitutive nuclear localisation[25]. We also found two key nonsynonymous base substitutions in the Ar hinge region and LBD, which are highly conserved among spiny-rayed fish (Acanthomorpha) Aras, including medaka and cichlid Aras but not in Japanese eel Ara[30]. Such substitution in the hinge region has changed the molecular property of Ara from a ligand-dependent- to a constitutive-nuclear localisation protein, while the substitution in the LBD has increased its transactivation capacity[30]. This suggests that retention of the two Ar ohnologs with neofunctionalisation and/or subfunctionalisation in the Acanthomorpha lineage can significantly contribute to reproductive diversification. A recent genetic study in the African cichlid (*Astatotilapia burtoni*) revealed that the two Ar ohnologs have diverged in their functional roles for male sexual characteristics[32]. However, the molecular basis of how the two Ar ohnologs exhibit functional divergence has not yet been elucidated.

To understand the molecular properties of Ar ohnologs in the diversification of sexual characteristics, we focused on the role of Ar in

Japanese medaka (*Oryzias latipes*), which is a well-defined model system for the study of sex determination[33] and sexual differentiation studies on morphological and behavioural traits[10,34–38]. Medaka has a male heterogametic (XX/XY) system, in which *dmy/dmrt1bY* on the Y chromosome determines their sexes[33]. Importantly, they show prominent external morphologies specific to males, such as modification of the anal fin with papillary processes and formation of a fork in the dorsal fin[39]. These sexual characteristics can be induced by androgen administration[40] and are probably regulated by ligand-dependent transcriptional regulation of Ars[36]. Males perform courtship behaviours consisting of a sequence of stereotyped actions that are easily quantified[35,38,41]. The sequence begins with the male approaching and following the female closely. The male then performs a courtship display, in which he swims quickly in a circular pattern in front of the female. If the female accepts the male, the male grasps her with his fins (termed "wrapping"), and they quiver together until eggs and sperm are released ("spawning"). If the male is not accepted, she either rapidly moves away from the male or assumes a rejection posture[38,41]. Furthermore, previous studies demonstrated that Ars are expressed in the brain[42] and can control the sexually dimorphic expression of neuropeptides and biologically active nonapeptides[10,43]. Although these findings indicate that Ars are involved in a broad range of male-specific traits in medaka, the differential role of the two Ars remains unclear. This study aimed to understand how evolutionary differentiation of Ar ohnologs contributed to the development of sexual characteristics that increase male fitness in teleosts, which provides insights into the mechanisms of radiation of teleost fishes driven by sexual selection.

In this work, we isolated medaka *ara* and *arb* mutants that had lost their ligand-dependent transactivation capacity from a medaka TILLING (targeting-induced local lesions in genomes) library and then comprehensively characterised the phenotypes of sexual characteristics in the single *ar* mutants (*ara* KO or *arb* KO) and double knockout mutants (*ar* DKO). We find that both *ara* and *arb* are required in males to achieve efficient reproductive success; *ara* and *arb* are functionally redundant in the regulation of courtship display but have divergent roles in other morphological and behavioural sexual characteristics. Unexpectedly, both *ar* ohnologs are not essential for spermatogenesis in medaka, which contrasts with the case of mouse *Ar* and may have permitted the *ar* evolution in teleosts by reducing part of the functional constraints on the gene(s). We also demonstrate that differences in transcriptional regulation and intracellular localisation could account for the divergent roles of *ar* ohnologs in the development of the sexual characteristics of teleosts.

## Results
### Screening of *ara* and *arb* knockout mutants (KOs)
By screening the medaka TILLING library, we identified founders possessing nonsense mutations in exon 6 of *ara* (S507X) and exon 4 of *arb* (L503X) (Supplementary Fig. 2a–c). These nonsense mutations lead to expression of the truncated protein products lacking the LBD that contains a key amino acid-substitution responsible for distinct transactivation responses of Ara and Arb in vitro[30] (Supplementary Fig. 2). These mutant alleles failed to mediate 11KT-induced transcription of androgen response element (ARE) reporter genes in COS7 cells, indicating that these mutations in *ara* and *arb* resulted in the loss of ligand-induced transactivation capacity (Statistical differences were assessed using two-sided Mann-Whitney U test (R version 4.2.0). *$P < 0.05$ (WT vs Ara mutant: $W = 16$, $P = 0.02857$; WT vs Arb mutant: $W = 16$, $P = 0.02857$, Supplementary Fig. 2d). Therefore, we established *ara*$^{-/-}$ (*ara* KO) and *arb*$^{-/-}$ (*arb* KO) medaka strains carrying homozygous null mutations of *ara* and *arb*, respectively, by crossing these founders. Furthermore, a double homozygous mutant strain (*ara*$^{-/-}$; *arb*$^{-/-}$, i.e. *ar* DKO) was established by crossing *ara* KO and *arb* KO strains.

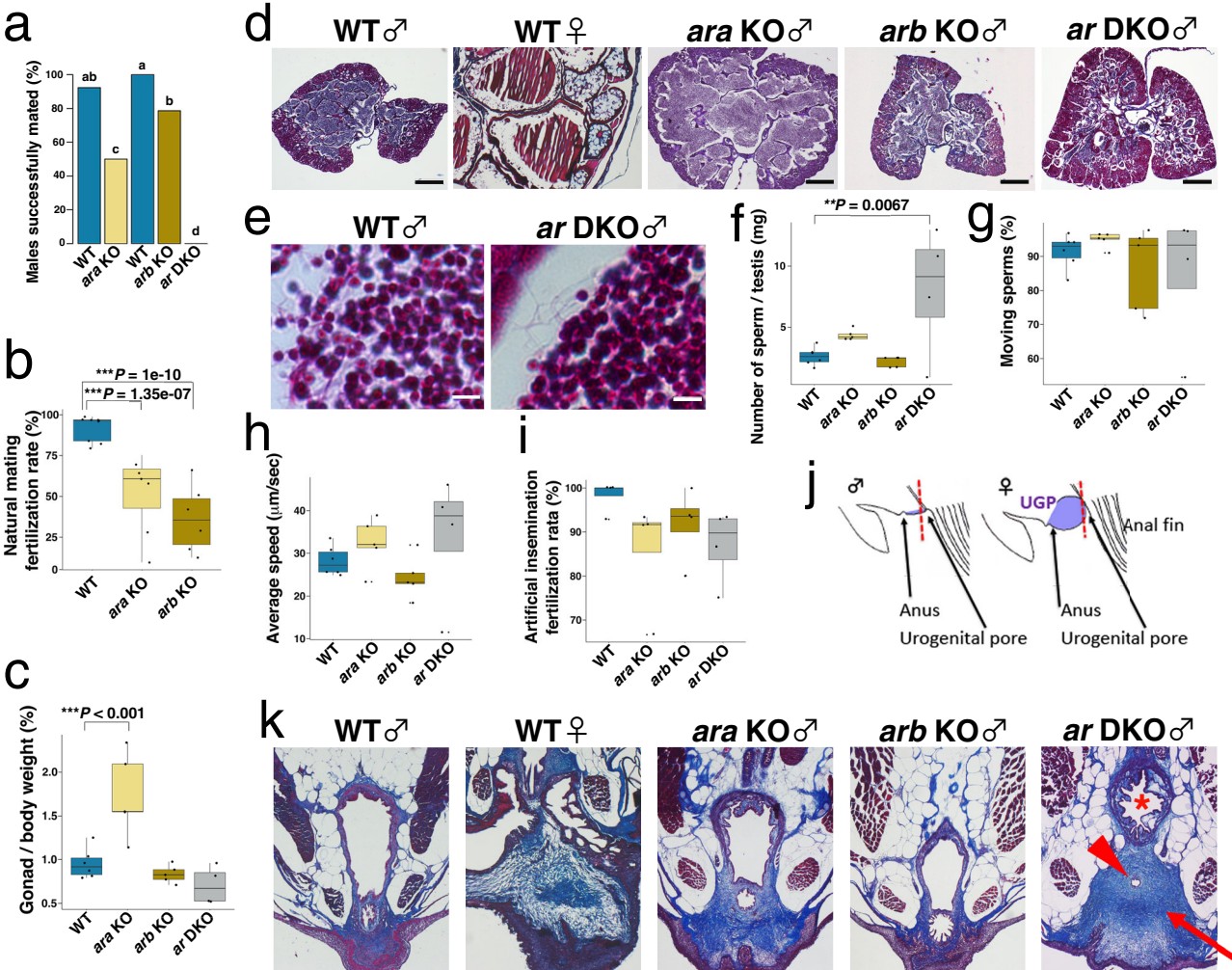

**Fig. 1 | Deficiency of either *ara* or *arb* reduces the frequency of spawning but maintains normal spermatogenesis. a** Frequency of mating tests in which a WT female laid eggs within 30 min after initiation of the mating. We used WT ($n = 11$, total 60 matings) and *ara* KO males ($n = 14$, total 76 matings) of the same litter, and WT ($n = 6$, total 42 matings) and *arb* KO ($n = 6$, total 41 matings) of the same litter, and *ar* DKO males ($n = 10$, total 43 matings), as mating partners for the WT females. Details of mating tests utilised for this and following behavioural analyses were shown in Supplementary Table 2. Means are shown. Different letters above the boxes indicate significantly different groups ($P < 0.05$, two-sided Fisher's exact test with Bonferroni correction: see Supplementary Table 3 for the statistical data.). **b** Percentage of fertilisation of ovulated oocytes from WT females in natural mating with WT, *ara* KO, and *arb* KO males ($n = 9$, 7, and 6, respectively). The *P* values of fertilisation rates for natural mating (**b**) and artificial insemination (**i**) are calculated using two-sided Dunnett's multiple comparison in a binomial GLMM (***$P < 0.001$). **c** Gonad/body weight ratios of adult WT, *ara* KO, *arb* KO, and *ar* DKO males ($n = 6$, 5, 5, and 4, respectively). The *P* values are calculated using two-sided Dunnett's multiple comparison test. **d** Representative micrographs of Masson/trichrome-stained sections of adult gonads of WT males, WT females, *ara* KO males, *arb* KO males, and *ar* DKO males ($n \geq 3$ for each genotype). Scale bars represent 200 μm. **e** Higher magnification of Masson/trichrome-stained sections of WT testis and *ar* DKO testis. Scale bars represent 5 μm. **f** Number of sperm per testis of adult WT, *ara* KO, *arb* KO, and *ar* DKO males ($n = 6$, 5, 5, and 4, respectively). The *P* values are calculated using two-sided Dunnett's multiple comparison test (**f**–**h**), **$P < 0.01$). **g** Percentage of sperm moving from WT, *ara* KO, *arb* KO, and *ar* DKO males ($n = 6$, 5, 5, and 4, respectively). **h** Average speeds of sperm movement from WT, *ara* KO, *arb* KO, and *ar* DKO males ($n = 6$, 5, 5, and 4, respectively). **i** Percentage of in vitro fertilisation of eggs from WT females using cryo-preserved sperm of WT, *ara* KO, *arb* KO, and *ar* DKO males ($n = 4$ for each genotype). **j** Line drawing of a lateral view (anterior to the left) of the medaka urogenital region. The blue colour indicates the region of urogenital papillae (UGP) that prominently develops in the female. The red dotted line shows the approximate levels of the sections in **k**. **k** Representative micrographs of Masson/trichrome-stained sections of the urogenital region of WT males, WT females, *ara* KO males, *arb* KO males, and *ar* DKO males ($n \geq 3$ for each genotype). Scale bars represent 200 μm. The urethra, sperm duct, and enlarged medulla surrounding the sperm duct are indicated by *, red arrowhead, and red arrow, respectively. In the box-plots, the centre line indicates the median, box limits indicate the upper and lower quartiles, the whiskers indicate 1.5× interquartile range, and the points are outliers. The blue, light brown, dark brown, and grey colours indicate WT, *ara* KO, *arb* KO, and *ar* DKO, respectively. Source data and statistical data are provided as a Source Data file and Supplementary Table 3, respectively.

## Roles of the two Ars in testicular development and spermatogenesis

To evaluate the influence of the Ar mutations on sex determination, we analysed the genetic sex by the amplification of *dmy* and then confirmed the gonadal sex by dissecting their gonads. All wild-type (WT) and mutant fish with a Y chromosome (XY chromosomes) had testes ($n = 10$ in each genotype), indicating that Ar mutations did not cause sex reversal. To examine the role of Ar mutations in male fecundity, we bred *ar* KO males with WT females. Although both *ara* KO males and *arb* KO males were fertile, their frequencies of successful reproduction and average rates of fertilisation were significantly decreased (Fig. 1a, b). No *ar* DKO males successfully bred with females under natural mating (Fig. 1a). In contrast, all mutant females were fertile and displayed fertilisation rates similar to those

**Table 1 | LC/MS analysis of androgen and oestrogen in testes**

|  | 11KT (ng/g) | T (ng/g) | E2 (ng/g) |
|---|---|---|---|
| WT | 10.07 ± 4.16 | 9.44 ± 3.75 | 1.13 ± 0.41 |
| *ara* KO | 25.39 ± 8.60 | 13.44 ± 1.69 | 0.41 ± 0.14 |
| *arb* KO | 16.34 ± 4.09 | 10.42 ± 2.57 | 0.23 ± 0.07[a] |
| *ar* DKO | 75.49 ± 5.05*** | 6.37 ± 1.27 | 0.54 ± 0.34[a] |

Data are mean (ng/g) ± s.e., *n* = 3, *** *P* < 0.001 (two-sided Dunnett's multiple comparison test, 11KT (WT vs *ar* DKO): estimate ± s.e. = 65.417 ± 8.170, *t* = 8.007, *P* = < 0.001).

[a]Indicates the value lower than minimum limit of determination.

Source data, the other statistical data, and chromatograms of 11KT and T in LC/M analysis are provided as a Source Data file, Supplementary Table 3, and Supplementary Fig. 14, respectively.

of WT females (Supplementary Fig. 3). We hypothesised that the lack of successful mating in *ar* DKO males was caused by defects in testicular differentiation and spermatogenesis, as demonstrated in previous studies in AR KO mice[1]. However, we discovered that the Ar mutants showed no difference in testicular morphology, except the *ara* KO males exhibiting hypertrophic testes filled with mature sperm, which correlated with their higher gonad/body weight ratio (gonadal-somatic index; GSI) (Fig. 1c, d). We found that even in *ar* DKO males, the testes contained mature sperms with tails in the seminiferous tubules (Fig. 1e).

Sperm quality was examined using a computer-assisted sperm analysis system (CASA). Total sperm number was significantly increased in *ar* DKO males (two-sided Dunnett's multiple comparison test: estimate ± s.e. = 5.4161 ± 1.5049, *t* = 3.599, *P* = 0.0067) and tended to increase in *ara* KO males (estimate ± s.e. = 1.7474 ± 1.4118, *t* = 1.238, *P* = 0.4944 (Fig. 1f), while there were no significant differences in the frequency of moving sperm and average speed of sperm in all mutant strains (Fig. 1g, h). Furthermore, no significant difference in the success rates of in vitro fertilisation using the cryopreserved sperms was found in all mutant strains (Fig. 1i), These results indicate that all the mutant strains produce functional sperms that were capable of fertilising the eggs. Quantification of testicular levels of 11KT and T by liquid chromatography-mass spectrometry (LC/MS) revealed the *ar* genotype significantly affected the 11KT level (one-way analysis of variance (ANOVA), Df = 3, *F* = 26.58, *P* = 0.000164) but not the T and E2 levels (one-way ANOVA, T: Df = 3, *F* = 1.352, *P* = 0.325; E2: Df = 3, *F* = 2.007, *P* = 0.192. The two-sided Dunnett's multiple comparison test indicates that the 11KT levels was significantly increased in the *ar* DKO males (estimate ± s.e. = 65.417 ± 8.170, *t* = 8.007, *P* < 0.001) but not the *ar* single mutant males (*ara* KO: estimate ± s.e. = 15.317 ± 8.170, *t* = 1.875, *P* = 0.218; *arb* KO: estimate ± s.e. = 6.267 ± 8.170, *t* = 0.767, *P* = 0.788) (Table 1). Expression analysis of gonad-specific genes, *vasa*, *P450c17* and *gsdf*, indicates that germ cells, Leydig cells, and Sertoli cells were formed in the testes of *ar* DKO as in *ar* single KOs (Supplementary Fig. 4).

Next, to investigate whether there were any morphological defects in tissues involved in sperm release, we observed the tissue structure of the urethra, sperm duct, and medulla of urogenital papillae in histological sections of the cloaca of the Ar KO strains (Fig. 1j, k). We found that the *ara* KO and *ar* DKO males showed a narrowed sperm duct cavity surrounded by medulla containing dense collagen fibres intensely stained by aniline blue, while the *arb* KO males showed a milder phenotype; the medulla developed in the posterior end of the sperm duct near the opening of the digestive tract (Supplementary Fig. 5). These results indicate that sperm duct constriction by the enlarged medulla may cause an increased number of sperm in the testes of *ara* KO and *ar* DKO males.

## The differential roles of the two Ars in the development of sexual characteristics

The functional contributions of *ara* and *arb* to the development of external sexual characteristics were examined in *ar* KO strains

(Fig. 2a–f). The *ara* KO males showed prominent masculine sexual characteristics in median fin morphology, including the formation of papillary processes that develop as branched bone nodules derived from anal fin rays, elongated anal fin rays (Fig. 2b, c), and forked dorsal fin with elongated rays (Fig. 2d), as with the WT males. In contrast, *arb* KO and *ar* DKO males showed shorter anal fin rays without any papillary processes, as observed in WT females (Fig. 2b, c). Quantitative assessment of the anal fin confirmed that papillary processes were lacking in the *arb* KO and *ar* DKO males but not in the *ara* KO males (Fig. 2e). These results were consistent with changes in the expression level of *lef1*, an androgen effector gene that is important for papillary process development[36], which was significantly reduced in the anal fins of *ar* DKO males but not in those of *ara* KO males (two-sided Dunnett's multiple comparison test, DKO: estimate ± s.e. = −14.915 ± 3.671, *t* = −4.062, *P* = 0.00931, *ara* KO: estimate ± s.e. = −4.013 ± 3.671, *t* = −1.093, *P* = 0.58592, arb: estimate ± s.e.= −9.84 ± 3.671, *t* = −2.68, *P* = 0.06707. Supplementary Fig. 6). We further quantified the outgrowth of anal fin rays by calculating the ratios of the length of the anterior 3rd and posterior 2nd fin rays to the width of the anal fin which was significantly larger in males than that in females of the WT (Fig. 2f). Such sexual differences in the relative length of the fin rays were observed in the both fin rays of *ara* KO and the posterior 2nd fin rays of the *arb* KO (two-sided Tukey's multiple comparison test, *ara* KO anterior 3rd: estimate ± s.e. = −10.9001 ± 2.1411, *t* = −5.091, *P* < 0.001, *ara* KO posterior 2nd: estimate ± s.e. = −14.8789 ± 2.1825, *t* = −6.817, *P* < 0.001; *arb* KO posterior 2nd: estimate ± s.e.= −9.2562 ± 2.1825, *t* = −4.241, *P* = 0.00202), but not in the anterior 3rd of the *arb* KO (1.2227 ± 2.1411, *t* = 0.571, *P* = 0.99909) and completely absent in both fin rays of the *ar* DKO males (anterior 3rd: estimate ± s.e.= 2.2646 ± 2.1411, *t* = 1.058, *P* = 0.96279, posterior 2nd: estimate ± s.e. = −0.1568 ± 2.1825, *t* = −0.072, *P* = 1 (Fig. 2f). Similarly, male-specific fork formation in the dorsal fin was observed in *ara* KO, in part, found in the *arb* KO but completely absent in the *ar* DKO males (Fig. 2d). These results indicate that *arb* predominantly controls masculine characteristics in the median fins of medaka.

We also examined another male sexual characteristic, the androgen-inducible increase in white pigment cells (leucophores) distributed from the nasal sac to the dorsal side of the eyes[44]. Gross observations showed that leucophores were decreased in the *arb* KO and *ar* DKO males but not in the *ara* KO males (Fig. 3), indicating that *arb* plays a critical role in the sexual differentiation of leucophore patterns.

Sexual dimorphism in tooth morphology such as larger lateral teeth of the upper and lower jaws in males[45] is another trait regulated by androgen in medaka[46]. We observed broken teeth in males, but not in females (Fig. 4a), which might be damaged by male-male competition. In fact, when two males and one female were placed in the same tank, a male showed aggressive competition and attacked the lateral body of another male using its face (Supplementary Movie 1). Micron-scale computed tomography (micro-CT) and bone staining revealed that the *ar* DKO males did not show any enlarged lateral teeth, as in WT females (Fig. 4a–c). Interestingly, while enlargement of the lateral teeth of the upper jaw was observed in the *arb* KO males as in WT males, less enlargement was detected in the *ara* KO males (indicated by arrows in Fig. 4a–c). Quantitative measurement of the teeth length by calculating the ratio of the length of the largest tooth in the upper jaw to the width of the upper jaw revealed that the teeth length was significantly decreased in the *ara* KO males compared to the WT males (two-sided Tukey's multiple comparison test, WT vs *ara* KO: estimate ± s.e. = −0.07172 ± 0.02614, *t* = −2.744, *P* = 0.0327), but not in the *arb* KO males (WT vs *arb* KO: estimate ± s.e. = −0.02544 ± 0.0242, *t* = −1.051, *P* = 0.5544) (Fig. 4d). These results indicate that masculinisation of tooth morphology is predominantly regulated by *ara*.

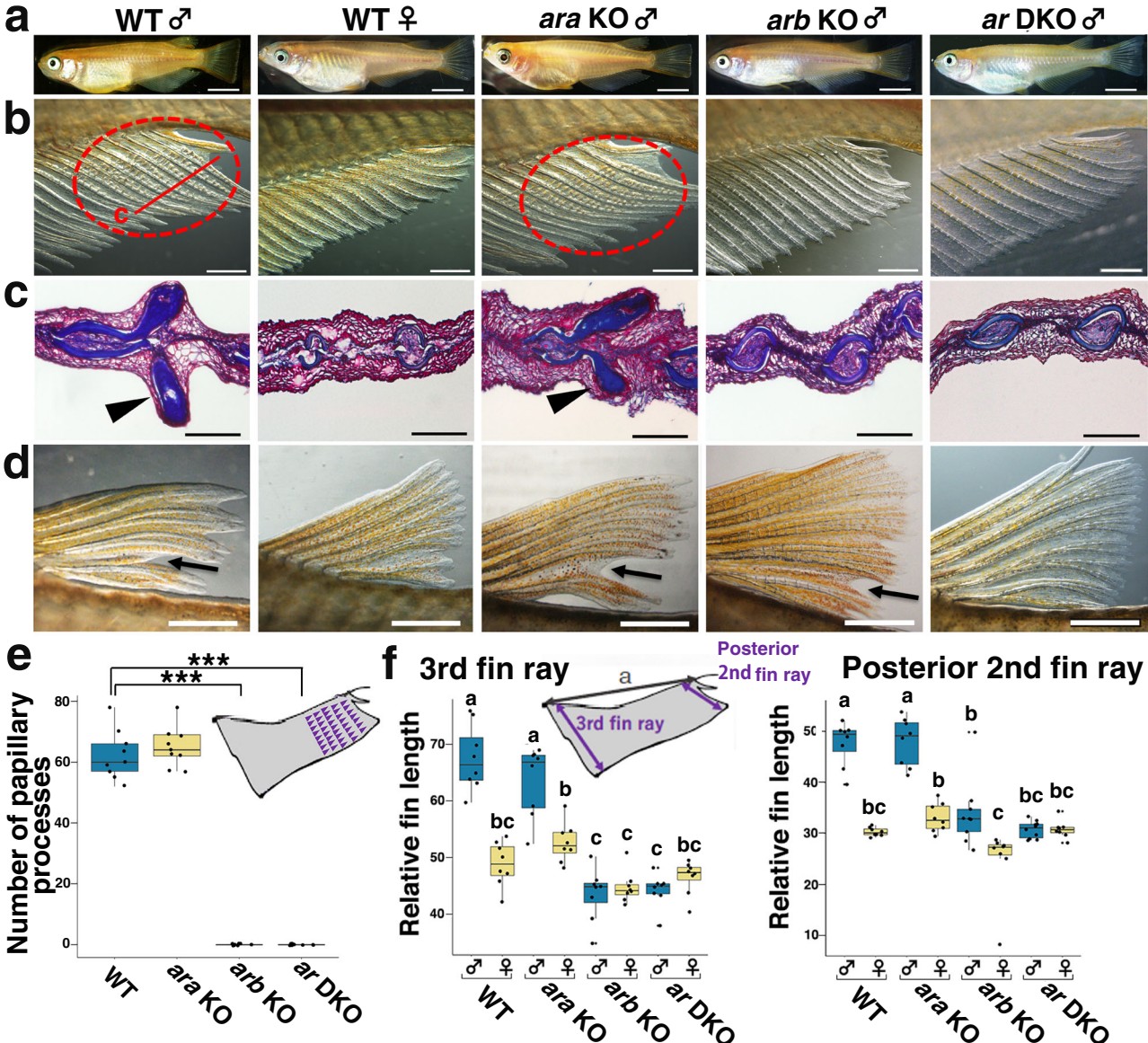

**Fig. 2 | *arb* KO males and *ar* DKO males display demasculinised fin structure, whereas *ara* KO males exhibit normal sexual characteristics in fin morphogenesis. a** Representative picture of the whole body of WT male, WT female, *ara* KO male, *arb* KO male, and *ar* DKO males ($n \geq 6$ of each genotype). Higher magnification of the anal fin (**b**), the micrographs of Masson/trichrome-stained sections of anal fin (**c**), higher magnification of the dorsal fin (**d**) of WT males, WT females, *ara* KO males, *arb* KO males, and *ar* DKO males, respectively ($n \geq 6$ of each genotype). The papillary processes developments were marked by red dotted circles in **b** and arrowheads in **c**. The plane of tissue section of anal fin is indicated by red line in **b**. Arrows in **d** indicate forks in the dorsal fin. **e** Number of developing papillary processes in adult WT, *ara* KO, *arb* KO, and *ar* DKO males ($n = 9$ for each genotype). The blue and light brown colours indicate WT and *ara* KO, respectively. The one-way ANOVA followed by two-sided Dunnett's multiple comparison test indicates that the number of papillary processes was not decreased in *ara* KO (estimate ±

s.e. = 2.667 ± 2.458, $t = 1.085$, $P = 0.572$) but completely absent in *arb* KO (estimate ± s.e. = −62.333 ± 2.458, $t = −25.36$, $P < 1e-04$) and *ar* DKO (estimate ± s.e. = −62.333 ± 2.458, $t = −25.36$, $P < 1e-04$). \*\*\*$P < 0.001$. **f** Lengths of the anterior 3rd and posterior 2nd anal fin rays relative to the anterior–posterior width (indicated as "a" in the picture) of the anal fin in WT, *ara* KO, *arb* KO, and *ar* DKO males and females ($n = 8$ for males and females of each genotype). The blue and light brown colours indicate males and females, respectively. Statistical differences were assessed using the two-way ANOVA followed by the Tukey–Kramer test. Different letters above the boxes indicate significantly different groups ($P < 0.05$, see Supplementary Table 3 for the statistical results). Scale bars in **a**, **b**, **c**, and **d** indicate 5.0 mm, 1.0 mm, 50 μm, and 1.0 mm, respectively. In the box-plots, the centre line indicates the median, box limits indicate the upper and lower quartiles, the whiskers indicate 1.5× interquartile range, and the points are outliers. Source data and statistical data are provided as a Source Data file and Supplementary Table 3, respectively.

## Differential effects of the two Ars on mating behaviours

Previous studies have shown that *ar* expression in the brain and the pituitary regulates the expression of several hormones that can control the reproduction and social behaviours in medaka[10,43,47]. Hence, we observed the mating behaviour of a test male (homozygotes for the WT, *ara* KO, *arb* KO, or *ar* DKO alleles) with a WT female (Supplementary Movies 2–6, short movies showing representative cases for each genotype). The *ar* DKO males mostly abolished courtship displays

during behavioural testing, whereas the single mutant males displayed courtships in over 90% of the tests (Fig. 5a), indicating that either of the two *ar*s is sufficient for this mating behaviour. However, we found differential effects of the two *ar*s on the frequency of courtship displays before spawning, which is an index of male sexual motivation (Fig. 5b). Although generalised linear mixed models (GLMM) showed no significant change in the frequency of courtship displays of the *ara* KO males (estimate ± s.e. = 0.1520 ± 0.1570, $z = 0.968$, $P = 0.3419$), the

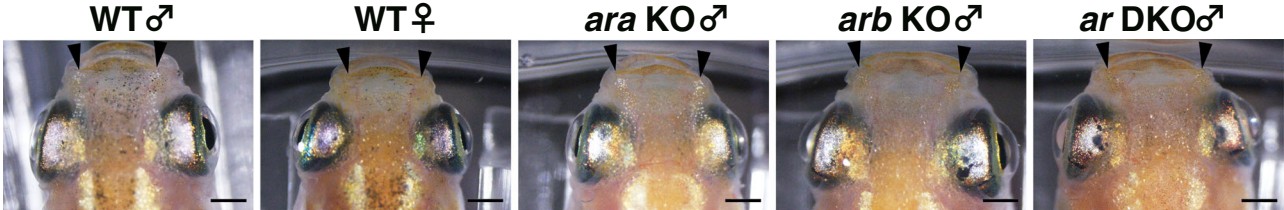

**Fig. 3 | *arb* KO males and *ar DKO* males exhibited a diminished number of leucophores from nasal sacs to the dorsal side of the eyes, whereas *ara* KO males exhibit normal leucophore differentiation.** Representative images of the upper jaw to the head of WT males, WT females, *ara* KO males, *arb* KO males, and *ar* DKO males (*n* = 5 for each genotype). Arrowheads indicate nasal sacs. Scale bars indicate 1 mm.

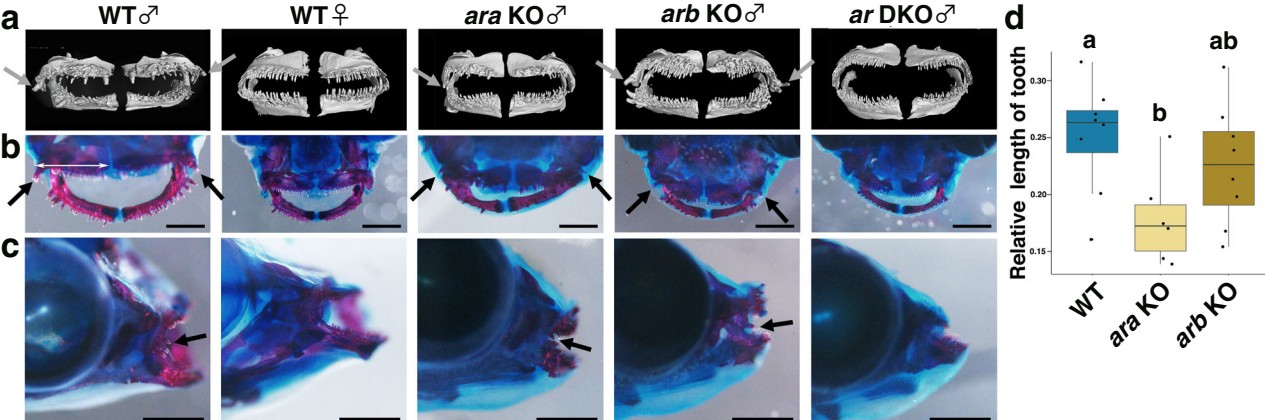

**Fig. 4 | *ar DKO* males have female-like teeth and *ara* KO males have a shorter lateral most tooth in the upper jaw, whereas *arb* KO males exhibit normal sexual characteristics in tooth morphogenesis. a** Micro-CT images of the teeth (frontal view). **b, c** Representative images of bone staining of the teeth (*n* = 9 for each genotype). Arrow indicates the most lateral tooth. Scale bars indicate 1 mm. **d** Lengths of the most lateral tooth relative to the lateral to medial width of the upper jaw stained with eosin (indicated as a line in **b**) in WT, *ara* KO, and *arb* KO males (*n* = 8, 6, and 8, respectively). Statistical differences were assessed using one-way ANOVA, followed by two-sided Tukey's multiple comparison test. The *ar* genotype significantly influenced the length of the tooth (one-way ANOVA, Df= 2, *F* = 3.796, *P* = 0.041). Two-sided Tukey's multiple comparison test indicates that the length of the tooth was significantly decreased in the *ara* KO males (estimate ± s.e. = −0.07172 ± 0.02614, *t* = −2.744, *P* = 0.0327). Different letters above the boxes indicate significantly different groups (*P* < 0.05). The blue, light brown, and dark brown colours indicate WT, *ara* KO and *arb* KO, respectively. In the box-plots, the centre line indicates the median, box limits indicate the upper and lower quartiles, the whiskers indicate 1.5× interquartile range, and the points are outliers. Source data and the other statistical data are provided as a Source Data file and Supplementary Table 3, respectively.

*arb* KO males showed significantly decreased frequency (estimate ± s.e. = −0.7641 ± 0.3297, *z* = −2.317, *P* = 0.03268) (Fig. 5b). This difference indicates that the *arb* KO males have lower sexual motivation than the *ara* KO and WT males.

A previous study demonstrated that medaka females prefer males with longer median fins[48]; therefore, we hypothesised that feminised fin morphology caused by *arb* deficiency but not by *ara* could affect female mate preferences. In contrast to our expectations, we found that both *ara* KO and *arb* KO males required significantly longer mating latency (*ara* KO, GLMM, estimate ± s.e. = 0.5081 ± 0.2342, *t* = 2.17, *P* = 0.03163; *arb* KO, GLMM, estimate ± s.e. = 1.1429 ± 0.2645, *t* = 4.321, *P* = 0.001353) (Fig. 5c). Consistently, the total number of wrapping rejections by the female was significantly increased in the case of both the *ara* KO (GLMM, estimate ± s.e. = 2.3981 ± 0.3685, *z* = 6.507, *P* = 4.32e-07) and *arb* KO males (GLMM, estimate ± s.e. = 1.6222 ± 0.4585, *z* = 3.538, *P* = 0.002858) (Fig. 5d). These results indicate that deficiency of either of the two *ar* genes decreases mate preference by the females. Furthermore, to understand the behavioural roles of the papillary processes that are thought to enable males to rub and prevent females from escaping during wrapping for spawning[41], we measured the duration of each wrapping event followed by egg spawning (Fig. 5e). Linear mixed models (LMMs) showed that the duration time was significantly reduced in both the *ara* KO (estimate ± s.e. = −7.853 ± 2.147, *t* = −3.658, *P* = 0.001675) and *arb* KO males

(estimate ± s.e. = −3.455 + 1.723, *t* = −2.005, *P* = 0.04403) (Fig. 5e), indicating that not only fin morphology but also any defects caused by *ara* deficiency affected wrapping behaviour.

Finally, to verify whether these behavioural defects caused by *ar* mutations decrease male reproductive success, we performed a mate choice test using two males—a WT male and an *ar* KO male—with a WT female. We found that both *ara* KO ($\chi^2$ = 23.458, df = 1, *P* = 1.277e-06) and *arb* KO males ($\chi^2$ = 12.224, df = 1, *P* = 0.0004718) successfully mated with females in significantly fewer trials compared to WT males (Supplementary Fig. 7), suggesting that both *ara* and *arb* are required for high reproductive fitness in males.

**Brain transcriptomic changes in *ar* KOs**

Because *ar* DKO males mostly abolished courtship displays, we performed RNA-seq analysis using whole brain tissues, including the pituitary, isolated from WT and *ar* DKO males to identify genes that regulate male courtship display. We identified 290 genes that were differentially expressed between the WT and *ar* DKO males. However, Gene Ontology (GO) enrichment analysis did not reveal significant enrichment of these genes in particular biological processes such as the regulation of social and reproductive behaviours (Supplementary Fig. 8). Identification of genes regulating the male courtship display would require identification of neurons that control the male reproductive behaviour and high-resolution gene expression analysis in such neurons.

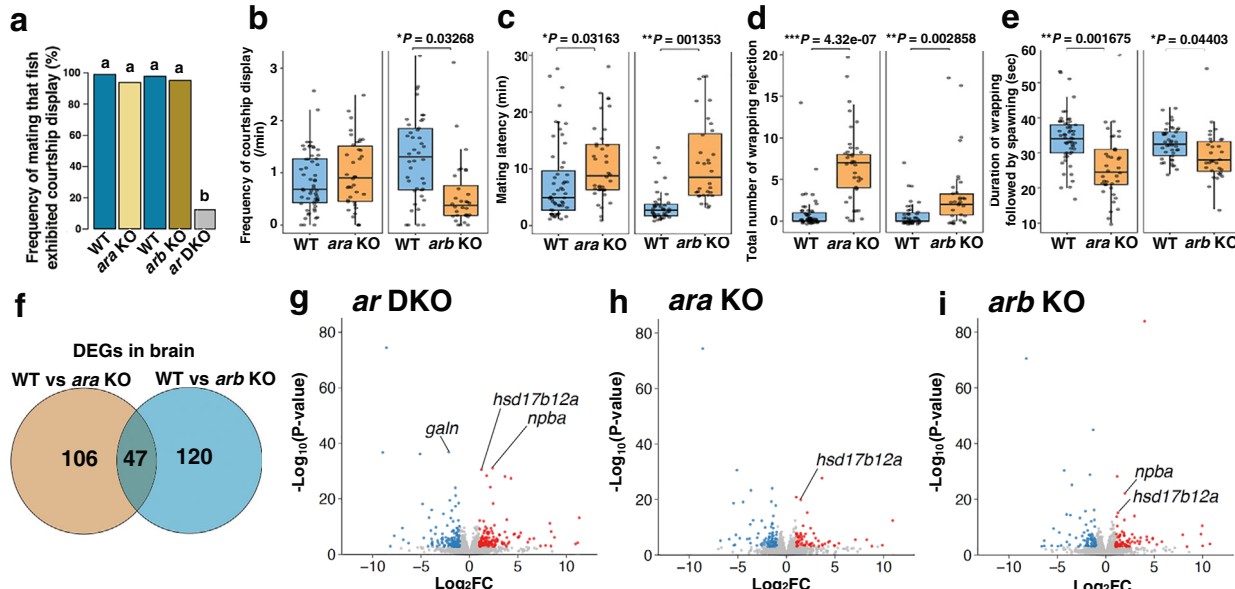

**Fig. 5 | Both *ara* and *arb* are required for high male reproductive success.**
**a** Percentage of mating tests with different females in which a male exhibited courtship display within a 30 min test period. We used WT (*n* = 11, total 59 matings) and *ara* KO males (*n* = 14, total 76 matings) of the same litter, and WT (*n* = 6, total 40 matings) and *arb* KO males (*n* = 6, total 40 matings) of the same litter, and *ar* DKO males (*n* = 10, total 43 matings) as mating partners for the WT females. The blue, light brown, dark brown, and grey colours indicate WT, *ara* KO, *arb* KO, and *ar* DKO, respectively. Means are shown. Different letters above the boxes indicate significantly different groups (*P* < 0.05, two-sided fisher's exact test with a Bonferroni correction). **b** Frequency of courtship displays before spawning. **c** Time required for spawning (mating latency). **d** Total number of wrapping rejections by female. **e** Duration of wrapping followed by spawning. For **b** to **e**, we used WT (*n* = 11, total 55 mating) and *ara* KO males (*n* = 13, total 38 matings) of the same litter and WT (*n* = 6, total 42 matings) and *arb* KO males (*n* = 6, total 32 matings) of the same litter. The blue and brown colours in panel **b** to **e** indicate WT and *ar* KO males,

respectively. Statistical differences were assessed by likelihood ratio test with two-sided alternative hypothesis in GLMMs (**b**–**d**) and LMMs (**b**–**e**) *\*P* < 0.05, \*\**P* < 0.01, \*\*\**P* < 0.001; T: WT vs *ara* KO, estimate ± s.e. = In the box-plots, the centre line indicates the median, box limits indicate the upper and lower quartiles, the whiskers indicate 1.5× interquartile range, and the points are outliers. Source data and statistical data are provided as a Source Data file and Supplementary Table 3, respectively. **f** RNA-seq of the whole brain with the pituitary in males. Venn diagram indicating overlap among inter-genotype differentially expressed genes (DEGs) (genes with >2-fold change in expression, false discovery rate (FDR) < 0.05). **g**–**i** Volcano plot with the log$_2$ fold-change on the *x*-axis and −log$_{10}$ of the *p* value on the y-axis in expression between WT and *ar* DKO males (**g**), WT and *ara* KO males (**h**), and WT and *arb* KO males (**i**). Each dot represents a single gene with differently expressed genes (DEGs)coloured (FDR < 0.05). The *p* values shown in the panels were calculated using two-sided Fisher's exact test with no adjustments for the multiple comparisons.

## Table 2 | LC/MS analysis of androgen and oestrogen in brains

|        | 11KT (ng/g)       | T (pg/g)          | E2 (pg/g)        | E1 (pg/g)         |
|--------|-------------------|-------------------|------------------|-------------------|
| WT     | 0.48 ± 0.11       | 235.47 ± 49.69    | 189.03 ± 28.62   | 287.00 ± 35.14    |
| *ara* KO | 1.86 ± 0.21\*\*   | 402.50 ± 22.83\*  | 168.87 ± 20.35   | 332.77 ± 28.69    |
| *arb* KO | 0.51 ± 0.12       | 178.27 ± 34.07    | 94.33 ± 29.47\*  | 159.43 ± 39.44\*  |
| *ar* DKO | 8.86 ± 0.34\*\*\* | 555.80 ± 25.07\*\*\* | 92.47 ± 4.65\*   | 217.93 ± 7.01     |

Data are mean ± s.e., *n* = 3. * *P* < 0.05, ** *P* < 0.01, *** *P* < 0.001, (two-sided Dunnett's multiple comparison test, 11KT: WT vs *ara* KO, estimate ± s.e. = 1.373 ± 0.307, *t* = 4.474, *P* = 0.00524; WT vs *ar* DKO, estimate ± s.e. = 8.38 ± 0.307, *t* = 27.3, *P* < 0.001; T: WT vs *ara* KO, estimate ± s.e. = 167.03 ± 48.89, *t* = 3.417, *P* = 0.0225; WT vs *ar* DKO, estimate ± s.e. = 320.33 ± 48.89, *t* = 6.553, *P* < 0.001; E2: WT vs *arb* KO, estimate ± s.e. = −94.7 ± 32.59, *t* = −2.906, *P* = 0.0479; WT vs *ar* DKO, estimate ± s.e. = −95.57 ± 32.59, *t* = −2.963, *P* = 0.0442; E1: WT vs *arb* KO, estimate ± s.e. = −127.57 ± 42.8, *t* = −2.981, *P* = 0.0431). Source data, the other statistical data, and chromatograms of 11KT and T in LC/M analysis are provided as a Source Data file, Supplementary Table 3, and Supplementary Fig. 15, respectively.

Next, we compared RNA-seq data from whole brain tissues of WT, *ara* KO, and *arb* KO males to identify genes responsible for behavioural changes in *ar* KOs. We identified 153 and 167 genes that were differentially expressed in *ara* KO and *arb* KO males, respectively, compared to WT males (Fig. 5f). Although GO enrichment analysis did not reveal the significant enrichment of these genes in particular biological processes (Supplementary Figs. 9 and 10), 106 of 153 and 120 of 167 genes were identified to be differentially expressed between *ara* KO and *arb* KO males, respectively (Fig. 5f). Among these genes, expression level of neuropeptide B a (*npba*),

which is known to regulate the female reproductive behaviour[49], was 5.17 and 3.98 times higher in the *ar* DKO and *arb* KO males compared to WT males, respectively (Fig. 5g, i). Additionally, expression level of *hsd17b12a*, whose product catalyses the transformation of estrone (E1) into E2[50] and 11-ketoandrostenedione (11KA4) to 11KT[51], was significantly higher in the males of all *ar* KO strains compared to WT males (Fig. 5g–i). We detected no up-regulation of *ara* in the *arb* KO and that of *arb* in the *ara* KO, which indicates that no detectable level of genetic compensation occurred between *ara* and *arb* at the whole-brain level. The quantification of steroid hormones by LC/MS revealed that the *ar* genotype significantly influenced the 11KT, T, E2, and E1 levels in the brains (one-way ANOVA, 11KT: Df = 3, *F* = 340.9, *P* = 8.90E-09; T: Df = 3, *F* = 24.42, *P* = 0.000222; E2: Df = 3, *F* = 4.724, *P* = 0.0352; E1: Df = 3, *F* = 6.351, *P* = 0.0164). The two-sided Dunnett's multiple comparison test indicates that the 11KT and T levels were significantly increased in the *ara* KO and *ar* DKO males (11KT for *ara* KO, estimate ± s.e. = 1.373 ± 0.307, *t* = 4.474, *P* = 0.00524; 11KT for *ar* DKO, estimate ± s.e. = 8.38 ± 0.307, *t* = 27.3, *P* < 0.001; T for *ara* KO, estimate ± s.e. = 167.03 ± 48.89, *t* = 3.417, *P* = 0.0225; T for *ar* DKO, estimate ± s.e. = 320.33 ± 48.89, *t* = 6.553, *P* < 0.001) but not the *arb* KO males (11KT, estimate ± s.e. = 0.03 ± 0.307, *t* = 0.098, *P* = 0.99928; T, estimate ± s.e. = −57.2 ± 48.89, *t* = −1.17, *P* = 0.5393), whereas the E2 level was rather decreased in the *arb* KO and *ar* DKO males (*arb* KO, estimate ± s.e. = −94.7 ± 32.59, *t* = −2.906, *P* = 0.0479; *ar* DKO, estimate ± s.e. = −95.57 ± 32.59, *t* = −2.963, *P* = 0.0442) (Table 2). These results indicate that *arb* predominantly represses the *npba* expression without inhibiting the E2 synthesis, whereas both *ar* ohnologs

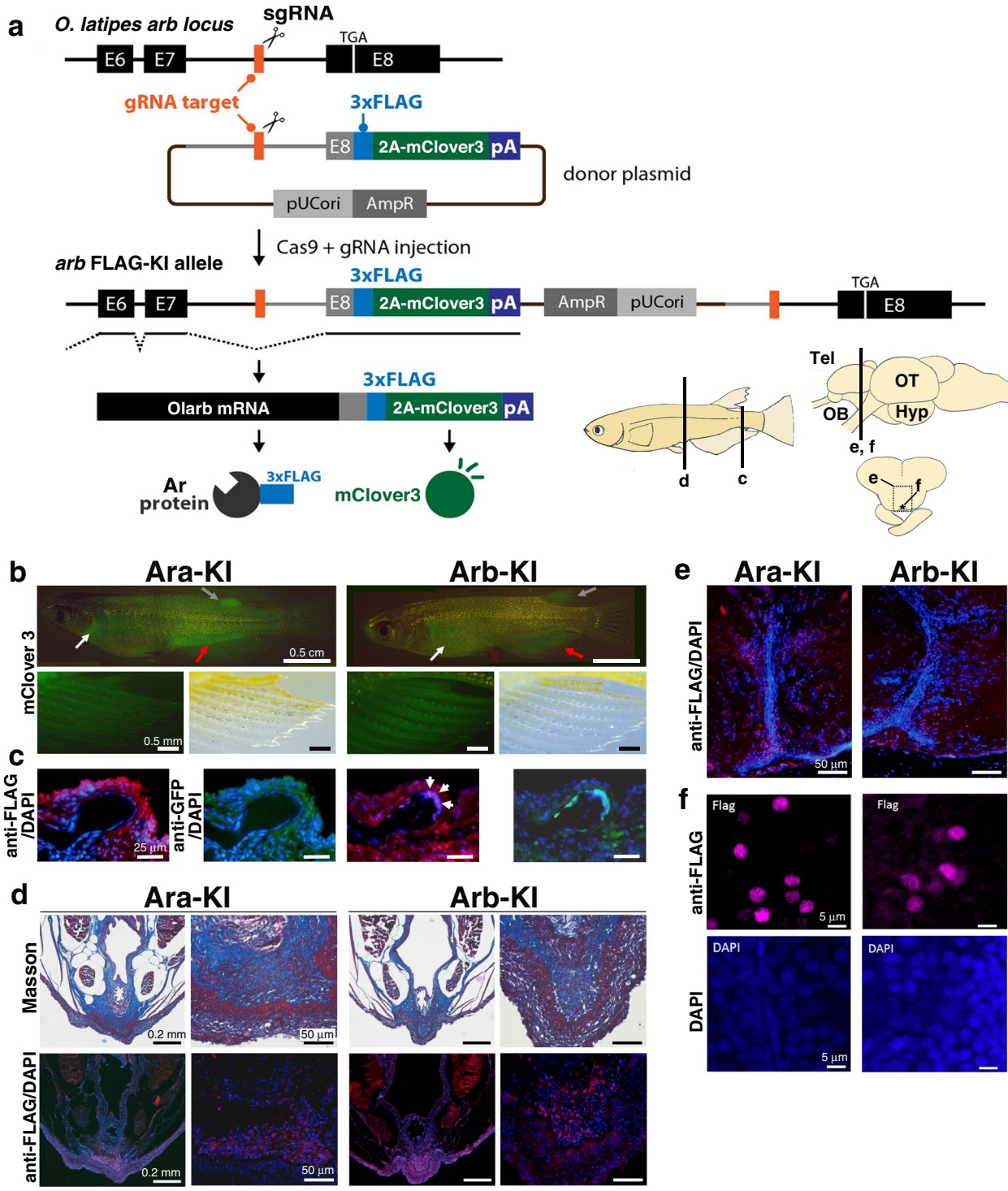

are required for repressing *hsd17b12a* expression in males, but *ara* is more effective in supressing 11KT and T synthesis. This is likely the reason why 11KT level increased in *ara* KO and *ar* DKO males. Such distinct roles for *ar* ohnologs in the regulation of Ar-biased gene expression in the brain may reflect the subfunctionalisation and/or neofunctionalisation of these ohnologs after the genome duplication event. Comparative genome sequence analysis combined with available ATAC-seq data in medaka[52] shows that putative *cis*-regulatory sequences of *ara* and *arb*, respectively, were partially conserved in teleost species, but not between *ara* and *arb*, supporting their subfunctionalization and/or neofunctionalization (Supplementary Fig. 11).

## Differences in expression and intracellular localisation of the two Ars

To visualise expression patterns and intracellular localisation of Ara and Arb in vivo, we generated two knock-in medaka strains, *ara^FLAG-2A-mClover3^* (Ara-KI) and *arb^FLAG-2A-mClover3^* (Arb-KI) expressing two distinct proteins, an epitope (3xFLAG)-tagged Ar and a green fluorescent protein (mClover3), from the endogenous *ar* loci (Fig. 6a). The Ara-KI adult males showed weak green fluorescence throughout the trunk and fins, and strong fluorescence in the regions adjacent to the pectoral, dorsal and anal fins. In contrast, the Arb-KI adult males showed a more restricted pattern of green fluorescence that localised primarily to the pectoral, dorsal and anal fins (Fig. 6b). In the papillary

**Fig. 6 | Visualisation of Ara and Arb expression in vivo by generating Ar-knock-in (KI) medaka strains. a** Strategy for the generation of Ara and Arb KI medaka strains. We designed a gRNA targeting *ara* intron 8 and *arb* intron 7 (the last intron). In the donor plasmids, we cloned a genomic fragment beginning from the end of exon 8 of *ara* and exon 7 of *arb* and ending just before the stop codon in the last exon, exon 9 of *ara*, and exon 8 (shown as black closed boxes with 'E8') of *arb*, where the 3xFLAG sequences (shown as blue boxes) and a P2A (2A peptide from porcine teschovirus-1)-mClover3 cassette (shown as green boxes) was placed in the frame. Thus, endogenous Ara and Arb were expressed as FLAG fusion proteins. Both AR-FLAG and P2A-mClover3 were expected to be expressed under the control of the endogenous Ar promoter. To generate KI medaka, sgRNA (for genome digestion in the final intron), donor plasmid, and Cas9 mRNA were co-injected into one-cell-stage medaka embryos. After injection, concurrent cleavage of the targeted genomic locus and the donor plasmid resulted in the integration of donor plasmid DNA containing 3xFLAG-T2A-mClover3 by non-homologous end joining (NHEJ). The scheme shows the forward integration of 3xFLAG-T2A-mClover3. **b** Expression of mClover3 in adult males of the two knock-in medaka strains, *ara*^FLAG-2A-mClover3 (Ara-KI) and *arb*^FLAG-2A-mClover3 (Arb-KI). White, grey, and red arrows indicate the regions adjacent to pectoral, dorsal, and anal fins, respectively (Ara-KI).

White, grey, and red arrows indicate the pectoral, dorsal, and anal fins, respectively (Arb-KI). **c** Immunohistochemical detection of FLAG-tagged endogenous Ara and Arb in longitudinal sections of papillary processes of the anal fin (6 μm thickness). The merged images represent red fluorescence for immunostaining of FLAG (anti-DDDDK-tag mouse mAb monoclonal antibody), green fluorescence for immunostaining of mClover3 (anti-GFP D5.1XP rabbit mAb monoclonal antibody), and blue fluorescence for nuclear staining by DAPI. The medaka Arb-FLAG, but not Ara-FLAG, translocated into the nuclei of cells located in the distal tip of a bone nodule of papillary processes (marked by white arrows). **d** Representative micrographs of Masson/trichrome staining and immunohistochemical detection of FLAG-tagged endogenous Ara and Arb in adjacent sections of the urogenital region (8 μm thickness). Nuclear localisation of Ara-FLAG and Arb-FLAG was observed in the medulla ventral to the sperm duct, where *ar* DKO exhibited hyperplasia. **e, f** Representative micrographs showing immunohistochemical detection of FLAG-tagged endogenous Ara and Arb, and DAPI counterstaining in the brain (12 μm thickness). **f** shows a higher magnification of the POA. Nuclear localisation of both Ara-FLAG and Arb-FLAG was observed in the POA. *n* = 6 for Ara-KI and Arb-KI males, respectively.

processes of the anal fin, stronger green fluorescence was observed in Arb-KI than in Ara-KI males (Fig. 6b). These differences in the expression patterns of Ara and Arb appear to be consistent with the results obtained from the comparative analysis of their *cis*-regulatory sequences (Supplementary Fig. 11). Next, we focused on the expression of Ara and Arb proteins in the cells located at the distal region of a bone nodule of papillary processes because we previously identified the expression of androgen effector genes in these cells[36]. We visualised Ara and Arb proteins in tissue sections of the anal fin by immunohistochemistry using a green fluorescent protein (GFP) antibody to identify the Ar-expressing cells and an anti-FLAG antibody to verify the nuclear localisation of Ar required for its activation as a transcription factor[53]. Consistent with the KO phenotype, we observed prominent GFP signals and nuclear localisation of FLAG signals in these Arb-KI cells, but not in Ara-KI cells (Fig. 6c). These results suggest that the *arb*-specific gene expression and the nuclear localisation of *arb* protein in the papillary processes account for *arb* KO-specific defects in this tissue.

In the urogenital region, nuclear localisation of both Ara and Arb was observed in the medulla ventral to the sperm duct (Fig. 6d), where *ar* KOs exhibited prominent hyperplasia. In the preoptic area (POA), a brain region known to regulate a wide range of instinctive behaviours including mating behaviour[54], we detected intense nuclear localisation of both Ara and Arb (Fig. 6e, f).

## Discussion

In the present study, we analysed the divergent roles of the TSGD-ohnolog pair *ara* and *arb* in the expression of sexual characteristics in medaka. We demonstrated that *ar* DKO males mostly abolished courtship displays and lacked external sexual characteristics, such as masculinisation of fin morphology and pigment cell patterns, resulting in infertility. These findings indicate the essential roles of Ars in male reproduction. Interestingly, *ar* DKO medaka showed successful spermatogenesis and no decrease in the number of mature sperms. This is consistent with the previous finding that medaka males deficient in the sex steroid synthesis-related gene *P450c17* possess well-developed testes with many spermatozoa with fertilising ability[55]. Our results clearly showed that androgen signalling is dispensable for spermatogenesis in medaka. In contrast, impairment of spermatogenesis has been previously reported in Ar-deficient animals such as ARKO mice, in which spermatogenesis was arrested at the pachytene spermatocyte[1] and *ar* deficient zebrafish showing infertility due to defective spermatogenesis and/or release of sperm[56–58]. Additionally, the stimulatory effect of 11KT on spermatogenesis has been reported in cultured testes of Japanese eel,[22] which represents the earlier branching teleost

groups[59]. These findings indicate that the functional contribution of androgen signalling to spermatogenesis has been lost, specifically in the lineage leading to medaka. Ar-mediated androgen activity also contributes to the maintenance of normal ovarian function in mouse[1,60] and zebrafish[56–58]. In contrast, Ar signalling is dispensable for female fecundity in medaka. The loss of contribution of Ar signalling in gametogenesis in both sexes might have reduced the evolutionary constraints on the *ar* ohnologs and accelerated the acquisition of exaggerated male-specific traits in the lineage leading to medaka. In fact, the medaka species that are widely distributed throughout southern and southeast Asia exhibit diversification of male-specific sexual characteristics in their fin morphology, pigmentation and behaviours[35,38,39,41,61,62].

To understand the exclusive role of *ara* and *arb* in the development of sexual characteristics, we analysed the morphological and behavioural traits of *ara* KO and *arb* KO males. We demonstrated that *arb* KO males showed a decreased frequency of courtship displays before spawning, reflecting their lower sexual motivation than WT males. In addition to such behavioural abnormalities, *arb* KO males lack external sexual characteristics, such as outgrowth of fin rays and papillary process development in the anal fin, as in females. Because females prefer males with longer fins[48], and the papillary processes are thought to rub the female to induce spawning and prevent the females from escaping[41], the demasculinisation of anal fin structure in *arb* KO males may reduce reproductive success. We also observed reduced reproductive success of the *arb* KO males due to the shorter duration of wrapping and subsequent spawning and rejection by females. This is consistent with the lower fertility caused by ablation of the part of the dorsal or anal fin in males[63]. In contrast, *ara* KO males showed intact sexual characteristics in fin morphogenesis but had shorter teeth. Although *ara* KO males did not show a decrease in sexual motivation considering the frequency of courtship displays before spawning, their frequency of mating was lower than that of WT males. Moreover, *ara* KO males were frequently subjected to wrapping rejection by females and exhibited a shorter duration of wrapping and subsequent spawning. These results indicate that their reproductive behaviour appears less attractive than that of WT males. In fact, *ara* KO males required more time to spawn with females. The lower fertility of *ara* KO males may be due to behavioural abnormalities as well as constriction of the sperm duct. Taken together, we revealed the differential roles of *ar* ohnologs associated with the sexual characteristics of teleosts−*ara* predominantly regulates the masculinisation of teeth and the reproductive behaviour eliciting female receptivity while *arb* plays essential roles in male-specific fin morphogenesis and sexual motivation. We confirmed that at least the phenotypes in fin morphogenesis

and spermatogenesis in the *ar* TILLING KO medaka lines were reproduced by *ar* TALEN KO medaka lines (Supplementary Figs. 12 and 13), demonstrating that the observed phenotypes were not specific to the *ar* TILLING KO lines.

Loss of function of either *ara* or *arb* reduced male reproductive fitness, whereas double mutations of *ara* and *arb* resulted in more severe phenotypes, indicating their overlapping and specific roles. The emergence of duplicated genes with overlapping roles that allow organisms to be phenotypically stable may relax selection pressure and enable innovation[64–67]. A recent study on African cichlid fish revealed that *ara* and *arb* diverged in their functions, wherein *ara* coded for reproductive and aggressive behaviours and *arb* for dominant bright colouration[32]. Corroborating this study, our findings suggest that external sexual characteristics, such as fin morphology and pigmentation patterns, are likely to be predominantly regulated by *arb* in both species. However, in contrast to the cichlid, the medaka *ara* regulates not only the performance of reproductive behaviour but also external sexual characteristics such as teeth enlargement that can be used in intra-male competition and the composition of medullary tissue in the cloaca, suggesting that the functional contribution of *ara* varies among species. A previous study comparing sequence differences between *ara* and *arb* showed that *ara* evolved 3.45 times faster than *arb* in medaka[30]. Taken together, these observations suggest that *arb* has been biased to maintain ancestral function under the stronger pressure of sexual selection compared to *ara* that has functionally diverged in each species. Further studies of other teleosts with two *ar* ohnologs and basal non-teleost ray-finned fishes (e.g. Polypterus) possessing a single *ar* gene are required to definitively determine the functions of *ar* ohnologs that have been derived from the ancestral gene and those that have been derived from innovated *ar* after the TSGD. So far, our previous analysis indicates that the Elopomorpha lineage including Japanese eel that represents the earlier branching teleost groups[59,68] have *ar* genes derived from ancestral genes before functional diversification, because the Japanese eel Ara and Arb did not show significant difference in their protein property in vitro[30] and have pleiotropic expression in various tissues such as testis, muscle, and spleen[69]. These observations suggest that the functional diversification of *ar* ohnologs has occurred in teleosts after the divergence of Elopomorpha lineage.

The generation of epitope-tagged AR-KI medaka lines enabled us to analyse the accurate expression pattern of each Ar ohnolog in vivo. We found prominent expression of Arb, but not Ara, in the distal region of a bone nodule of papillary processes of the anal fin. In addition, we observed the localisation of Arb, but not Ara, in the cellular nuclei of this region. Such differential expression patterns of Ar ohnologs at the tissue and cellular levels may explain the loss of papillary processes in *arb* KO males. We also observed the cytoplasmic localisation of Ara in the anal fin, which appears to be inconsistent with a previous study showing constitutive nuclear localisation of Ara in COS-7 cells[25]. This may be explained by tissue- or cell type-specific expression of nuclear localisation signal (NLS)-binding proteins such as importin family proteins because Ara-specific amino acid substitution has been identified in its NLS that contacts the importin α proteins[30]. We observed the nuclear localisation of both Ara and Arb in other cells, such as the POA neurons of the brain and the medulla of the urogenital region. Together, these findings indicate that the differences in the regulation of both the transcription and intracellular localisation of Ars can become possible determinants of the functional differences of Ar ohnologs generated by TSGD.

Identification of the downstream effector genes of Ars will enable us to reveal the genetic changes that underlie phenotypic novelty. Previous studies in mice have identified genes important for signalling pathways during developmental processes, including sonic hedgehog (*shh*), bone morphogenetic protein (*bmp*), and genes of the Wnt/β-catenin pathway[70,71]. Interestingly, we observed abnormal activation of *neuropeptide B a* (*npba*) and *hsd17b12a* expression, which were 5.17 and 4.12 times higher, respectively, in the brains of *ar* DKO males than in those of WT males (Fig. 5g). *npba* is known as a downstream target of E2/Esr2b, which is necessary for female-specific mating behaviour[49]. Esr2b/E2 signalling plays a decisive role in the suppression of male-typical courtship display in females[38]. Besides, the expression of *hsd17b12a* has been shown to catalyse the transformation of E1 into E2[50]. Therefore, we expected Esr2b signalling to be activated by increasing E2 levels in the brains of *ar* DKO males. However, contrary to our expectations, *ar* DKO males did not show any increase in brain E2 levels, indicating that the abnormal activation of *npba* expression in *ar* DKO males is not due to the rise in brain E2 levels and is maybe regulated by Ar signalling. Although we have not yet identified the downstream effector genes that account for each Ar-ohnolog-specific behaviour, we found that Ara and Arb have distinct repertoires of downstream effector genes in the brain. Abnormal activation of *npba* was exclusively found in *arb* KO males but not in *ara* KO males, suggesting that Arb has opposing effects on the expression of the female-biased gene necessary for female mating behaviour. Arb and Esr2b may interact competitively for the regulation of downstream effector genes. We aim to identify such genes in the future by conducting gene expression analysis in nerve nuclei with higher resolution and chromatin immunoprecipitation sequencing analysis of their target genes using the epitope-tagged AR-KI medaka lines generated in this study.

In conclusion, we showed that in medaka, both *ar* ohnologs are not required for spermatogenesis but are required for high male reproductive success because of their overlapping and specific functions in the development of other male-specific morphological and behavioural sexual characteristics (Fig. 7). These findings suggest that the functional diversification of *ar* ohnologs has been promoted in the teleost lineage in two different ways—the loss of their roles in spermatogenesis and the gene duplication that has likely resolved the pleiotropic function derived from their ancestral gene. Furthermore, the generation of epitope-tagged AR-KI medaka lines enabled us to analyse the transcriptional regulation and intracellular localisation of each Ar, which filled a gap in knowledge, that is, the mechanism by which Ar ohnologs exhibit functional differences in vivo. Our results provide a comprehensive foundation for understanding the mechanisms underlying the diversification of sexual characteristics driven by *ar* gene duplication in teleosts.

## Methods

### Animals
All the procedures and protocols were approved by the Institutional Animal Care and Use Committee of the National Institute for Basic Biology (15A005, 14A003, 13A023, 12A020, 11A028) and Kyushu University (A21-043-0, A19-137-0, A19-137-1, A19-137-2, A29-088-0, A29-088-1, A29-088-2). Japanese medaka (*Oryzias latipes*) were bred and maintained under artificial reproductive conditions with 14 and 10 h of light (8:00–22:00) and dark cycles at 26–28 °C. They were fed commercial pellet food (Hikari Lab; Kyorin Co., Ltd., Hyogo, Japan) 2–3 times a day. We used sexually matured (4–10 months old) male and female medaka producing fertilised eggs for at least three consecutive days until the day before the experiment. The *ar* DKO males were also reared with females in the same tank for one week until the day before the experiment. The gloss morphology and pigment pattern of the adult fish were observed using a stereo fluorescent microscope FLUO III™ (Leica, Wetzlar, Germany). Micrographs were taken using a digital charge-coupled device camera DP-73 (Olympus, Tokyo, Japan).

### Screening for *ara* and *arb* KOs from the medaka TILLING library
In teleost fishes, two distinct subtypes of Ars were first identified as ARα and ARβ in Nile tilapia (*Oreochromis niloticus*), Japanese eel, and

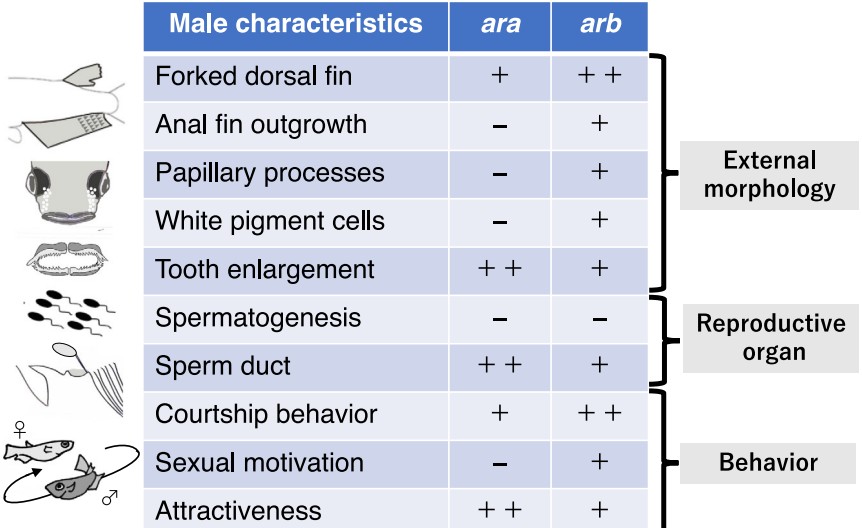

| Male characteristics | *ara* | *arb* | |
|---|:---:|:---:|---|
| Forked dorsal fin | + | + + | External morphology |
| Anal fin outgrowth | – | + | |
| Papillary processes | – | + | |
| White pigment cells | – | + | |
| Tooth enlargement | + + | + | |
| Spermatogenesis | – | – | Reproductive organ |
| Sperm duct | + + | + | |
| Courtship behavior | + | + + | Behavior |
| Sexual motivation | – | + | |
| Attractiveness | + + | + | |

**Fig. 7 | Summary of the contribution of *ara* and *arb* to sexual characteristics.** Based on the phenotypes of *ara* KO and *arb* KO, the contribution of each of the *ar* ohnologs to male sexual characteristics is indicated by + or –. For the characteristics regulated by both, the *ar* ohnolog that predominantly controls them is indicated by ++.

Atlantic croaker (*Micropogonias undulatus*)[69,72,73]. We have then distinguished the medaka Ars, *ara* (ARα, Genbank accession: AB252233, Ensembl ID: ENSORLG00000008220, located on chromosome 10), and *arb* (ARβ, AB252679, Ensembl ID: ENSORLG00000009520, located on chromosome 14), based on phylogenetic analysis considering those first identified as ARα and ARβ in Nile tilapia and Japanese eel[25,69]. Both *ara* KO and *arb* KO medaka strains were established using the TILLING approach. Briefly, the medaka TILLING library consisting of frozen sperm and genomic DNA obtained from 5760 $F_1$ male fish[74] was screened for mutations in *ara* and *arb* by high resolution melting (HRM) analysis[75] of fragments amplified by genomic PCR using KOD-Plus (Toyobo, Osaka, Japan) and ResoLight Dye (Roche, Mannheim, Germany) on a LightScanner System (96 software v2.0, Idaho Technology Inc., Salt Lake City, UT, USA). The PCR conditions for *ara* KO screening were as follows: preheating at 95 °C for 2 min, followed by 45 cycles of denaturation at 95 °C for 10 s, annealing at 58 °C for 30 s, and elongation at 68 °C for 30 s, and finally heating at 68 °C for 2 min. The conditions for *arb* KO screening were the same except the annealing temperature was set at 56 °C. The primer sequences used for KO screening are listed in Supplementary Table 1. Fluorescence measurements were collected from 40 to 80 °C, and melting curves were analysed using the LightScanner System software version 2.0 (Idaho Technology Inc.). The PCR products selected as KO candidates were incubated with ExoSAP-IT reagent at 37 °C for 15 min, then inactivated at 80 °C for 15 min, and then directly sequenced on an ABI PRISM 3130xl Genetic Analyser (Life Technologies, Waltham, MA, USA) using a BigDye Terminator v3.1 cycle sequencing kit (Life Technologies). ID:5383 and ID:3793 were identified as founders harbouring nonsense mutations in the 6th exon of *ara* (*ara$^{S507X}$*) and the 4th exon of *arb* (*arb$^{L503X}$*) (Supplementary Fig. 2). $F_2$ heterozygotes of KO males were obtained by artificial insemination using the National BioResource Project (NBRP) Medaka. To remove background mutations, heterozygous KO males were backcrossed with WT females of OK-Cab (NBRP ID: MT830) until the $F_5$ generation. After repeated backcrossing, the strains were maintained by crossing heterozygous females and males to obtain WT and homozygous siblings for phenotypic analyses. The *ara* and *arb* double heterozygous males and females (*ara$^{+/-}$*; *arb$^{+/-}$*) were obtained by breeding *ara$^{-/-}$* males with *arb$^{-/-}$* females. The *ara* and *arb* double heterozygous males and females were crossed to obtain *ara$^{-/-}$*; *arb$^{+/-}$* males and females. The *ara* and *arb* double homozygous males and females (*ara$^{-/-}$*; *arb$^{-/-}$*) were obtained by breeding *ara$^{-/-}$*; *arb$^{+/-}$* males with *ara$^{-/-}$*; *arb$^{+/-}$* females. The genotype of each fish was determined by direct sequencing of the PCR product containing the mutant site from the fin clips of the adult fish using the same sets of primers for TILLING screening. The genetic sex of each fish was determined by genomic PCR according to a previous report[76].

### Transcriptional activity of the Ar KO
The expression vectors for medaka cDNAs for Ara (pCMV-medaka Ara$^+$) and Arb (pCMV-medaka Arb$^+$) were generated as previously described[30,77]. The mutations identified by TILLING were introduced into these expression vectors using the PrimeSTAR Max mutagenesis basal kit (Takara Bio, Shiga, Japan) and primers listed in Supplementary Table 1 to obtain pCMV-medaka Ara$^-$ and pCMV-medaka Arb$^-$.

COS-7 cells cultured in 24-multiwell plates ($5.0 \times 10^4$ cells/well) were transfected with 400 ng/well of the PRE/ARE tk Luc reporter plasmid[78], 0.8 ng/well of pRL-SV40 (*Renilla* luciferase vector) as the internal control, and 80 ng/well of the expression vector for Ars, using 1.5 μL/well of TransFast transfection reagent (Promega, Tokyo, Japan). After 6 h of transfection, the cells were incubated for 12 h in Dulbecco's modified Eagle's medium with 10% charcoal-treated foetal bovine serum, either in the presence or absence of $10^{-8}$ M 11KT (K-8250, Sigma-Aldrich, St. Louis, MO, USA). The reporter gene activities were determined using the dual-luciferase reporter assay system (Promega) with values normalised to pRL-induced activities (i.e. firefly luciferase activity/*Renilla* luciferase activity). Each assay was performed in triplicates and repeated four times independently. The statistical significance of the reporter gene assay was analysed using two-sided Mann-Whitney *U* test by R version 4.2.0.

### Measurement of sex steroid levels
Adult testes and whole brains of WT, *ara* KO, *arb* KO, and *ar* DKO males at ~6 months of age were collected 1 to 3 h after the onset of the light period, We judged that the 6 months old was proper to check the testicular function of medaka, because it has been shown that the plasma T and 11KT levels peak in male medaka at 8 months old[79]. 11KT, T, E1, and E2 were quantified in each testis or brain using liquid chromatography-tandem mass spectrometry (LC-MS/MS) at ASKA Pharma Medical Co. Ltd (Kanagawa, Japan) as described by Nishiike et al.[38]. Steroid levels are expressed as nanogrammes or picogrammes of

each steroid per gram of wet tissue weight. Statistical differences between each mutant and the WT males were analysed using one-way ANOVA followed by two-sided Dunnett's multiple comparison test in the 'glht' function of the R package *multcomp* version 1.4-20.

## Quantification of male fecundity

Males (WT, *ara* KO, *arb* KO, and *ar* DKO) at six months of age were separated from females in the evening the day before the experiment. After measuring the body weight, the testes were removed, weighed, and homogenised with forceps for 1 min in 66 µL of ice-cold cryopre-servation liquid (10% N,N-dimethylformamide) in foetal calf serum until the residual testis fragments became invisible. The obtained suspension was sucked into a glass capillary (Hirschmann disposable microcapillary, 10 µL/capillary, three capillaries for each testis) (Thermo Fisher Scientific, Waltham, MA, USA) and cryopreserved for artificial insemination according to the protocol provided by NBRP Medaka (https://shigen.nig.ac.jp/medaka/medakabook/index.php). Sperm suspension (4 µL) was mixed with 36 µL of ice-cold Iwamatsu's balanced salt solution (BSS) by pipetting and immediately loaded into the sperm counting chamber (SC12-01-C, Leja Products B. V., Niew-Vennep, Netherlands). Sperm motility, average speed, and sperm concentration were examined using CASA (SMAS3 (ver.3.1.11.357), DITECT, Tokyo, Japan). For artificial insemination, 10 µL of sperm suspension in a glass capillary was thawed in 100 µL of ice-cold BSS and immediately added to the unfertilised eggs according to the protocol provided by NBRP Medaka. The GSI was calculated as follows: GSI (%) = gonad weight × 100/body weight. For the statistical analysis of the fertilisation rates, two-sided Dunnett's multiple comparison in binomial GLMMs was conducted in R version 4.2.0 with the packages *lme4* version 1.1-31 and *multcomp* version 1.4-20. The statistical sig-nificances of gonad/body weight, sperm motility test were analysed using one-way ANOVA followed by two-sided Dunnett's multiple comparison test in the 'glht' function of the R package *multcomp* version 1.4-20.

## Generation of Ar-FLAG-mClover3 knock-in (KI) medaka strains

We generated KI medaka strains using a targeted gene trap approach mediated by the clustered regularly interspaced short palindromic repeats (CRISPR) and CRISPR-associated protein 9 (CRISPR/Cas9) system[80]. In this approach, the targeted insertion of a donor DNA vector harbouring a targeted genomic sequence can be induced by concurrent cleavage of the chromosomal target site and the circular donor DNA using the Cas9 ribonucleoprotein complex with a single guide RNA (gRNA) sequence. The donor vector for *ara* (pUC-Ara-3xFLAG-2A-mClover3 (a brighter and more photostable variant of green fluorescent protein)[81] includes a 484 bp segment of the partial genomic fragment consisting of a 322 bp segment of the 8th intron sequence and a 162 bp segment of the 9th exon sequence until just before the stop codon, while the vector for arb (pUC-Arb-3xFLAG-2A-mClover3) includes a 562 bp segment of the genomic fragment con-sisting of a 39 bp segment of the 7th exon sequence, 367 bp segment of the 7th intron sequence, and 156 bp segment of the 8th exon sequence. The partial genomic sequence of each vector was followed by in-frame insertion of a targeting cassette composed of a 3x FLAG sequence, P2A sequence derived from porcine teschovirus-1[82], the coding sequence of mClover3 from pNCS-mClover3 (Addgene Plasmid # 74236)[81], and SV40 poly A signal (Fig. 6a). Each donor vector was inserted into a gRNA-targeting site in the 8th or 7th intron of *ara* or *arb*, respectively. The resulting alleles could express mRNAs under the control of endo-genous *ar* promoters, and the mRNAs are expected to be translated into two distinct proteins, namely, the C-terminal FLAG-tagged Ar and mClover3, owing to a ribosomal skipping by the P2A sequence[82].

Cas9 mRNA and gRNAs were generated as previously described[7,83]. Briefly, Cas9 mRNA was transcribed from the pCS2 + hSpCas9 vector using an mMessage mMachine SP6 Kit (Thermo Fisher

Scientific). The sgRNAs were synthesised using the AmpliScribe T7-Flash Transcription Kit (Epicentre, Madison, WI, USA) with templates that were PCR-amplified using 57-mer oligonucleotides containing a T7 promoter sequence and a 20-mer sequence for a custom target (Supplementary Table 1). The donor plasmid vectors were extracted using an alkaline lysis miniprep approach, incubated at 55 °C for 30 min with 0.5% SDS and 0.8 µg/µL of proteinase K to eliminate the residual RNase activity, and then purified using NucleoSpin Gel and PCR Clean-up kit (MACHEREY-NAGEL) with the Buffer NTB.

Microinjection of the OK-Cab strain into fertilised eggs was per-formed following a previously established method[84]. We injected ~1 nL of a mixture containing 100 ng/µL Cas9 mRNA, 10 ng/µL sgRNA, and 5 ng/µL donor plasmid. The targeted insertion of each donor vector in $F_1$ or later generations was examined by PCR genotyping using the primer pair GFP-127RV and ara-KI-5FW or arb-KI-5FW (Supplementary Table 1).

## Histological and immunohistochemical analyses

Medaka tissues were fixed in 4% paraformaldehyde in PBS at 4 °C. The fixed tissues were decalcified using decalcifying solution B (041–22031, Fuji Film Wako, Osaka, Japan) for 48 h, dehydrated in graded methanol, embedded in paraffin, and cut into 6-µm-thick sections for Masson/ trichrome staining and immunohistochemical analyses. Brain tissues were cut into 8-µm-thick sections and subjected to immunohisto-chemical analysis, as described previously[36]. Briefly, after antigen retrieval by incubating slides in 0.1 mM citrate buffer (pH 6.0) in an autoclave (121 °C) for 1 min, the sections were blocked for 1 h using 1xPBS containing 0.1% Tween 20, 2% BSA, and 2% foetal bovine serum, and incubated with anti-DDDDK-tag (FLAG) mouse mAb monoclonal antibody (M185-3L, MBL, Nagoya, Japan, 1:300 dilution) and anti-GFP D5.1XP rabbit mAb monoclonal antibody having cross-reactivity to the mClover3 (#2956, Cell Signaling, Danvers, MA, USA, 1:300 dilution) overnight at 4 °C. The sections were washed with 1xPBS with 0.1% Tween 20 (PBT), replaced with blocking buffer, and then incubated with Alexa 555-conjugated goat anti-mouse IgG(H + L), F(ab')₂ frag-ment (#4409, Cell Signaling, 1:300 dilution), and Alexa 488-conjugated anti-rabbit IgG(H + L), F(ab')₂ fragment (#4412, Cell Sig-naling, 1:300 dilution) for 1 h at room temperature. After counter-staining with the Cellstain DAPI solution (D523, Dojindo, Kumamoto, Japan), fluorescent signals were observed using an upright fluores-cence microscope (Axioplan 2; Zeiss, Jena, Germany) or a confocal laser scanning microscope (FV1000, Olympus).

GFP fluorescence in the whole body of adult fish was observed using a stereo fluorescent microscope FLUO III™ (Leica). Micrographs were taken using a digital charge-coupled device camera DP-73 (Olympus).

## Micro-computed tomography (micro-CT) and bone staining

For micro-CT, the head parts were fixed as described above and then scanned using a Phoenix nanotom m (Baker Hughes, Houston, TX) at the JMC Corporation (Yokohama, Japan). For skeletal stains, Alcian blue/Alizarin red staining was performed as previously described[85]. The lengths of the fin and tooth of each fish were calculated using Adobe Photoshop CC 2019 (ver. 20.0.2) using a picture taken with a digital charge-coupled device camera (DP-73, Olympus) using a stereo microscope FLUO III™ (Leica). The statistical analyses of the tooth length and the number of papillary processes were conducted one-way ANOVA followed by two-sided Dunnett's multiple comparison test. The statistical significances in the fin lengths were analysed using two-sided Tukey's multiple comparison test in the 'glht' function of the R package *multcomp* version 1.4-20.

## RNA isolation

Each male was anaesthetised using 0.84% tricaine methanesulfonate (Sigma-Aldrich), and then the anal fins were separated into distal parts

(distal three bone segments), anterior parts (the first to 10th fin rays counted from the anterior first fin ray), and posterior parts (the 11th fin ray to the posterior end where papillary processes generally develop on the fin rays). Total RNA was prepared from the posterior parts of the anal fin using an RNeasy Micro Kit (Qiagen, Hilden, Germany). WT, *ara* KO, and *arb* KO males were mated with a WT female and then anaesthetised immediately after courtship. *ar* DKO males were anaesthetised after 5 min of mating with a WT female. All matings were done within 1 h of the onset of the light period. The whole brain including the pituitary gland of each fish was collected from anaesthetised fish, and total RNA was purified using the RNeasy Plus Universal Kit (Qiagen).

## RNA-seq

RNA-seq of the whole brain along with the pituitary gland was performed at Azenta (Tokyo, Japan). Briefly, cDNA libraries were prepared using MGIEasy RNA Directional Library Prep Set V2.0 (MGI Tech, Beijing, China) with 500 ng of total RNA after selection with the NEB-Next Poly(A) mRNA Magnetic Isolation Module (New England Biolabs, Ipswich, MA, USA). The libraries were sequenced using the DNBSEQ-G400 platform (MGI Tech, Tokyo, Japan) with $2 \times 150$ bp paired-end reads.

After adaptor and quality trimming using Trim Galore 0.6.4_dev with Cutadapt 1.18, transcripts were quantified with RNA-seq reads mapped to the transcriptome sequences of *O. latipes* (Hd-rR) (Ensembl 105; https://www.ensembl.org/) using salmon v1.3.0. The transcript counts were converted to gene-level counts using the R package *tximport*, and statistical analysis was performed using the R package *edgeR* v3.34.1. Of the 19,227 genes retained after filtering genes with lower expression with the 'filterByExpr' function, genes showing >2-fold change with <5% false discovery rate (FDR) in Fisher's exact test were identified as differentially expressed genes between the WT and one of the Ar KO strains.

GO enrichment analysis was conducted using ShinyGO 0.75 (https://academic.oup.com/bioinformatics/article/36/8/2628/5688742) using the Ensembl gene IDs of the differentially expressed genes as input.

## Quantitative RT-PCR

First-strand cDNA was transcribed from 500 ng of total RNA using SuperScript III (Thermo Scientific). The relative RNA equivalents for each sample were determined by normalising their levels to those of the internal control gene *rpl7* [86]. Gene expression levels in the posterior part of anal fins of WT, *ara* KO, *arb* KO, and *ar* DKO males were quantified using 7500 real-time PCR with SYBR Green master mix (Thermo Scientific). The statistical significance of the differences was examined using one-way ANOVA followed by two-sided Dunnett's multiple comparison test in the 'glht' function of the R package *multcomp* version 1.4-20. Primer sets used for the quantitative RT-PCR (qRT-PCR) analyses of *lef1* and *rpl7* were designed by Applied Biosystems Primer Express 2.0.0 (Thermo Fisher Scientific) and listed in Supplementary Table 1.

## Behavioural assays

To observe 1-to-1 mating behaviour, a male and a female were transferred into a single acrylic tank (200 mm length × 75 mm width × 150 mm height and filled with 2 L of water at 26.5 °C) and separated by a white plastic partition in the evening (18:00–19:00) of the day before the assay. After the onset of the light period (8:30–10:00), the behavioural trial was initiated by removing the partition. Mating behaviour was recorded until spawning using a digital video camera HDR-PJ800 (Sony, Tokyo, Japan). To exclude the effect of familiarity between the examined male and female, each test fish was crossed with another individual until the day prior to the assay. If the female lay eggs within 30 min of a behavioural trial, we analysed the number of courtship

displays before spawning and the interval between the removal of the separator and successful egg spawning as mating latency, which is an indicator of female mate preference in medaka[7]. Wrapping behaviours that did not lead egg spawning were counted as wrapping rejections. If the wrapping led the spawning successfully, the period from the initiation of wrapping to the end of spawning was measured as a wrapping duration. We summarised the number of the mating tests for each analysis in Supplementary Table 2.

All videos in each dataset were analysed by a single viewer to ensure consistency. Statistical analysis for the frequency of reproduction and the frequency of mating that fish exhibited courtship display was conducted using two-sided fisher's exact test with a Bonferroni correction in R version 4.2.0. For the other behavioural data, statistical analysis was conducted using GLMMs and LMMs in R version 4.2.0 with the package *lme4* version 1.1.30. Poisson distributions with a log link function were used to analyse the number of courtship displays and wrapping rejections, whereas gamma distributions with a log link function were used for mating latency. The wrapping durations were analysed using LMMs. Genotypes and individual ID of males were included as fixed factors and random intercepts in each model. In the models for courtships, mating latency was included as a log-linear offset to standardise the number of events per unit time. The significant effect of the male genotype in each model was evaluated using the likelihood ratio test between the full and null models, excluding the fixed factor.

To directly measure the female mate preference to WT male or *ar* KO male, we performed round-robin tournaments using five WT males and five *ar* KO males, that is, a total of 25 combinations of mate choice test for each *ar* KO strain. The scores were given as the frequency with which each male mated with a female. Similar to the 1-to-1 mating behavioural analysis above, this mate choice test was initiated after the onset of the light period (8:30–10:00). A WT male and *an ara* KO or *arb* KO male were transferred to a single tank with a WT female. Then, their behavioural interactions were recorded until spawning. Each male individual was distinguished by the pigment pattern of the melanophores and fin shape. A male individual that led to egg spawning was judged as a successful mating partner under observation of each video recording. If the female did not lay eggs within 1 h of a behavioural trial, we judged the trial to be a draw. Effects of each *ar* mutation on the mate behaviour were statistically analysed using the chi-squared ($\chi^2$) test of independence in R version 4.2.0 with the package *lme4* version 1.1.30.

To measure the female fecundity of the *ar* KO strains, a female of WT, *ara* KO, *arb* KO, or *ar* DKO strains was transferred into a test tank with a WT male, and they were separated by a partition on the day before the test, as described above. After removing the separator in the morning of the test day, we analysed whether the females spawned the eggs. If the female successfully spawned, the total number of eggs obtained and their fertilisation rates were quantified. This test was repeated for 10 consecutive days by exchanging the males daily.

## Reporting summary

Further information on research design is available in the Nature Portfolio Reporting Summary linked to this article.

## Data availability

RNA-seq data for the whole brain with a pituitary gland used in this study are available from DDBJ under accession number DRA013672. The transcriptome sequences of *Oryzias latipes* (Hd-rR; ASM223467v1) in Ensembl Release 100 (http://dec2021.archive.ensembl.org/) were used for the RNA-seq analysis. The cDNA sequences for medaka *ar* ohnologs are available from Genbank (Accession number AB252233 for *ara*; AB252679 for *arb*). The genome sequences for medaka *ar* ohnologs are available from Ensembl (ID: ENSORLG00000008220 for *ara*; ENSORLG00000009520 for *arb*). The OK-Cab (NBRP ID: MT830),

the TILLING KO lines (ID: TA5383 for *ara* KO; TA3793 for *arb* KO), the AR-KI lines (TG1341 for Ara-KI; TG1342 for Arb-KI), and the TALEN KO lines (ID: Ara(del1) MT1560 for *ara* KO; Arb(del10) MT1561 for *arb* KO) are available from NBRP medaka. Representative examples of the behavioural tests are shown in Supplementary Movies 1–6. The full video data are available from the authors upon request. Source data are provided with this paper.

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

## Acknowledgements

We thank the National BioResource Project (NBRP) Medaka for providing the F$_2$ heterozygotes of TILLING mutant medaka strains, Cab strain, and the protocols for cryopreservation and artificial insemination of sperms. We thank Dr. Yoshihito Taniguchi for constructing the medaka TILLING library; Kayo Inaba, Hiroko Egashira, Tomoe Fujino, and Kazumi Sunny Tsukazawa for assistance in rearing the medaka; Azusa Yanogawa for assistance in sperm cryopreservation; Shinichi Chisada (Kyorin University), Yusuke Takehana (Nagahama Institute of Bio-Science and Technology), Hideaki Takeuchi (Tohoku University), Minoru Tanaka (Nagoya University), Hiroshi Akashi (Tokyo University of Science), Tohru Kobayashi (University of Shizuoka), Kenji Toyota (Kanazawa University), Hisayo Fukuda (Kansai Medical University), Takeshi Kitano (Kumamoto University), and Yoshitaka Nagahama (NIBB) for discussions and technical assistance. We thank the Center for Advanced Instrumental and Educational Support of the Faculty of Agriculture (Kyushu University) for technical support. This study was supported by the Ministry of Education, Culture, Sports, Science and Technology (MEXT) and the Japan Society for the Promotion of Science (JSPS, grant numbers JP15K07138, JP19K06741, JP20H04928, and JP16H06280 to Y.O. and JP19H03049 and JP22H00386 to K.O.). This work was also supported by the Astellas Foundation for Research on Metabolic Disorders, The Naito Foundation, NIBB Collaborative Research Program (17-317, 18-324, 19-309, 20-309, 21-203, 22NIBB309), the 2nd Women Researchers Promotion Program, AY2016 Support for childbirth and childcare in Women Researchers Promotion Program, Support for Women Returning from Maternity and Parental Leave programme in Kyushu University to Y.O., and grants from the Ministry of the Environment, Japan to T.I.

## Author contributions

Y.O. and T.I. conceived and designed the study. Y.O., S.A., M.Y., and Y. Katayama performed the experiments. Y.O., S.A., E.W., K. Okamoto, K. Okubo, and H.S. analysed the data. Y.Y., I.H., T.Y., A.K., Y. Kamei, K.N., S.M., T. Sato, G.Y., K. Okamoto, K. Ohta, T. Sakamoto and M.E.B contributed material/analytical tools. Y.O., S.A., H.O., and T.I. wrote the manuscript with inputs from other authors.

## Competing interests

The authors declare no competing interests.

## Additional information

[1]Laboratory of Aquatic Molecular Developmental Biology, Graduate School of Bioresource and Bioenvironmental Sciences, Kyushu University, Fukuoka 819-0395, Japan. [2]Center for Promotion of International Education and Research, Faculty of Agriculture, Kyushu University, Fukuoka 819-0395, Japan. [3]Graduate School of Life Sciences, Tohoku University, Sendai, Miyagi 980-8577, Japan. [4]Laboratory of Neurophysiology, National Institute for Basic Biology, Okazaki, Aichi 444-8787, Japan. [5]Department of Basic Biology, Graduate University for Advanced Studies (SOKENDAI), Hayama, Miura, Kanagawa 240-0193, Japan. [6]Center for Optical Research and Education (CORE), Utsunomiya University, Utsunomiya, Tochigi 321-8585, Japan. [7]Ushimado Marine Institute, Graduate School of Natural Science and Technology, Okayama University, Ushimado, Setouchi, Okayama 701-4303, Japan. [8]Department of Aquatic Bioscience, Graduate School of Agricultural and Life Sciences, The University of Tokyo, Bunkyo, Tokyo 113-8657, Japan. [9]Department of Physiology, Division of Life Sciences, Faculty of Medicine, Osaka Medical College, Takatsuki, Osaka 569-8686, Japan. [10]Laboratory of Bioresources, National Institute for Basic Biology, Okazaki, Aichi 444-8585, Japan. [11]Center of Interuniversity Bio-Backup Project, National Institute for Basic Biology, Okazaki, Aichi 444-8787, Japan. [12]Spectrography and Bioimaging Facility, National Institute for Basic Biology, Okazaki, Aichi 444-8585, Japan. [13]Laboratory of Marine Biology, Faculty of Agriculture, Kyushu University, Fukuoka 819-0395, Japan. [14]Amphibian Research Center/Graduate School of Integrated Sciences for Life, Hiroshima University, Higashi-Hiroshima, Hiroshima 739–8526, Japan. [15]Faculty of Advanced Engineering, Tokyo University of Science, Katsushika, Tokyo 125-8585, Japan. [16]Graduate School of Nanobioscience, Yokohama City University, Yokohama, Kanagawa 236-0027, Japan. [17]Department of Developmental Genetics, Institute of Advanced Medicine, Wakayama Medical University, Wakayama 641-8509, Japan. [18]Division of Nephrology-Hypertension, School of Medicine, University of California, San Diego, La Jolla, CA, USA. [19]Present address: Laboratory of Genome Editing Breeding, Graduate School of Agriculture, Kyoto University, Kyoto 606-8502, Japan. [20]Present address: Faculty of Marine Science and Technology, Fukui Prefectural University, Obama, Fukui 917-0003, Japan. ✉e-mail: ogino@agr.kyushu-u.ac.jp

