## [Peer Review File · Nature Communications]

Evolutionary differentiation of androgen receptor is responsible for sexual characteristic development in a teleost fishREVIEWER COMMENTS

Reviewer #1 (Remarks to the Author):

The manuscript sounds good, designated well with enough data in the result. However, some concerns should be addressed.

- 1. The authors used TILLING method to knock out ara and arb, while they used CRISP/Cas9 to knock in ara and arb, which showed that they could knock out genes with CRISP/Cas9 very efficiently. In this case, why did they not knock out ara and arb with CRISP/Cas9 to check whether the mutants produced by CRISP/Cas9 had the same phenotype as the mutants produced by TILLING.**
- 2. Both ara and arab mutants are fertile, while the ara and arab double mutants can not breed with females under natural mating. The authors should examine whether a genetic compensating response has occurred.**
- 3. In Fig.1, the pictures with only HE staining are not enough, in situ hybridization or immune-histochemistry of some gonad specific genes must be provided, such as germ cells gene vasa, Sertoli cells genes dmy, gsdf and Leydig cells gene to check why the ara and arab mutants reduce the frequency of spawning.**
- 4. Many pictures are not clear enough, such as Fig 1d, 1e and 1K, the authors should provide pictures with high resolutions.**
- 5. The ms should have more words about the knock in plasmids, such as what is mClover3, why the fish was in green?**
- 6. Fig legends of 1e are not in consistent with Fig.1e**
- 7. In the ms, "fertilisation" should be "fertilization"; "localization" should be "localization"; "Visualisation" should be "Visualization".**
- 8. Line 540, six month medaka are too old, they are not good for the experiment, since 3 month old medaka are matured.**
- 9. Full name of 'WT' should be given as it appears for the first time.**

Reviewer #2 (Remarks to the Author):

The current manuscript analysed in detail the unique and overlapping functions of two androgen receptor (ar) orthologs in various sexual characteristics in one of the teleost model species, the Japanese medaka. The authors used both mutants for either ar copies (ara KO and arb KO) and double mutants (ar DKO) to compare the effect resulting from the loss of either ar gene. A wide range of sexual characteristics were examined carefully, including mutant fecundity, morphology, mating success, behaviour and brain gene expression patterns. With these comparisons, the authors identified distinct functions of either ar ortholog. Furthermore, the authors used epitope-tagged AR-KI mutants show that the functional difference between the two orthologs potentially arise from their different transcriptional regulation and intracellular localisation. Overall, this manuscript is very logically structured and easy to read with clear writing. I find the experiments conducted and analyses performed thorough and convincing. I especially enjoyed the included video of aggressive competition among the male fish. I do not have major criticism on this manuscript, but have a few comments that I hope the authors can address to further improve the manuscript.

I couldn't find information regarding the levels of divergence between ara and arb. Do they share functional domains? Are they located in syntenic blocks in the genome? Do they have very divergent putative promoter sequences that might be underlying the different transcriptional regulation?

The authors included the discussion of identification of the downstream effector genes of Ars (Line420-line434), but I didn't find the relevant results for this investigation in the result section "Brain transcriptomic change in ar KOs". Maybe consider restructuring the presentation of these results and discussion.

More on this section, line295-line296, the authors wrote 'suggesting the occurrence of a small number of genes exhibiting low expression but having large effect on neuronal signal'. I find the support to this strong conclusion rather weakly supported, and it is not clear if it is based on the GO enrichment analysis or the fact that there are 290 DE

genes? Is 290 genes fewer than expected compared to other studies analysing gene expression change related to behaviour? Could it be that the brain gene expression would be more or less exaggerated based on the context? Maybe consider rephrasing? Line 190 and Line 191 present contradictory ideas, potential a typo?

Line 354- line 358: I'm not quite following how ar signalling resolves sexual conflict in medaka, could the author elaborate on this idea? 'Acquisition of exaggerated male-specific traits in the lineages leading to medaka' would be more convincing if the author comment on the level of sexual dimorphism in the lineages leading to medaka first.

Line 450-452: "resolved the pleiotropic function derived from their ancestral gene". I couldn't find relevant information in the manuscript for the function of ar in species that branched prior to TSGD to support this statement on 'ancestral' state.

Reviewer #3 (Remarks to the Author):

Overall feedback:

This manuscript describes a very thorough study in which the authors sought to identify the molecular properties of androgen receptor orthologs underpinning the diversification of sexual characteristics in a fish, the Japanese medaka (*Oryzias latipes*). As such, this paper provides important insights into the mechanisms of radiation of teleost fishes driven by sexual selection.

I enjoyed reading this article. It covers an important topic using a study species that is less conventional than, for example, rodents, which have been used extensively in studies of sexual dimorphism. I also appreciate that the authors tested for effects of deficiency of androgen receptors alpha and beta on a broad range of morphological, reproductive, and behavioural traits (e.g. fin size and shape, tooth enlargement, spermatogenesis, courtship behaviour).

Because of the high standard and thoroughness of the study and manuscript, I have quite few comments and recommend a minor revision. However, I do think that these points will help to improve the manuscript.

Line-by-line feedback:

Line 57: I am not sure about the use of the phrase 'behavioural attractiveness' here given that 'attractiveness' is perceived by the behavioural signal receiver, which depends on a whole host of other factors. Is there a more accurate term that could be substituted in here?

Line 62: No capital is needed for 'first'.

Lines 131-132: It would be useful here to just briefly describe the types of actions that are typical of male medaka reproductive behaviour.

Line 269-272: The term 'wrapping' has not yet been defined and so the reader will not have proper context for this statement. It would be useful to earlier define this term and, in addition, briefly describe the typical mating behaviour of male and female medaka.

Lines 682-683: 'The number of males escaping wrapping by the male was counted as the number of wrapping rejected by the female.' Again, these behaviours are not sufficiently described and so will be potentially confusing to readers.

Line 684-685: How were the behaviours recorded? Using a behaviour-logging software? If so, the name and version of the software should be provided. Also, was the observer blind to treatment while scoring trials?

Line 695: 'The mate choice test was performed using a round-robin tournament with five males.' It is not clear what this means. Please provide sufficient detail and clarity when describing all behavioural assays so that there is no chance of misinterpretation by readers.

Line 713: I recommend that all data and code, including for behavioural trial data, are made publicly available alongside the article. In particular, uploading all of the data that would be required to re-run your analysis is crucial to making your research transparent.

We thank the reviewers for their many thoughtful and helpful comments throughout the manuscript. Their major points were:

- 1) Additional experiments are required to check whether the phenotypes of the TILLING KO fish are consistent with those of KO fish generated by another reverse genetics method, such as CRISPR/Cas9 KO.
- 2) More detailed analysis of testicular differentiation, such as the *in situ* hybridization or immune-histochemistry of some gonad specific genes, is requested to check why the *ara* and *arb* KO males reduce the frequency of spawning.
- 3) Comparison of *cis*-regulatory sequences of *ara* and *arb* genes in teleost is requested. The *ara* and *arb* might be taken different transcriptional regulation. This possibility needs to be addressed.
- 4) More detail on the divergence of *ara* and *arb*, such as the genetic synteny, and domain structure of these proteins, are requested.

To address these issues, we have included the following changes in the revised manuscript:

- 1) We generated *ar* TALEN KO medaka lines carrying mutations leading the Ar protein truncation at its N-terminus. We confirmed that the resulting fish showed morphological phenotypes indistinguishable from those of the TILLING KO fish in their fin and testis.
- 2) We performed section *in situ* hybridization analysis of gonad specific genes, *vasa* (germ cell marker), *P450c17* (Leydig cell marker), and *gsdf* (Sertoli cell maker) in testes of WT, *ara* KO, *arb* KO, and *ar* DKO males. The results indicate that each type of cells successfully differentiated in the *ar* DKO as well as in the *ar* single KOs. These requested data are now included in the Supplementary Information.
- 3) We performed the comparative genome sequence analysis combined with the available ATAC-seq data in Medaka. The result shows that putative *cis*-regulatory sequences of *ara* and *arb*, respectively, were partially conserved in teleost species, but not between *ara* and *arb*, supporting their subfunctionalization and/or neofunctionalization.
- 4) The requested details of the domain structure of Ars, the similarity of deduced amino acid sequences of each domain of medaka Ara and Arb with human AR, and molecular phylogenetic tree of vertebrate Ar genes are now included in Supplementary Fig 1. We also included the genome synteny information in the main text.

The list of figures that we added in the revised manuscript

- **Supplementary Fig. 1. Phylogenetic analysis of Ar genes**
- **Supplementary Fig. 4. *In situ* hybridization for gene expression analysis of testis in wild-type and ar KO strains**
- **Supplementary Fig. 11. Comparison of *cis*-regulatory sequences of ar genes in teleost**
- **Supplementary Fig. 12. Design of transcription activator-like effector nucleases (TALENs) and frame-shift mutations in the ara and arb genes**
- **Supplementary Fig. 13. Fin and testicular phenotypes in ar TALEN KO medaka**

We have also addressed other minor points to improve the manuscript. Our detailed point-by-point responses to the reviewers' specific comments are set out below.

(The sentences that we changed are indicated by red and our responses to the comments from reviewers are shown as bold sentences.)

REVIEWER COMMENTS

Reviewer #1 (Remarks to the Author):

The manuscript sounds good, designated well with enough data in the result. However, some concerns should be addressed.

1. The authors used TILLING method to knock out ara and arb, while they used CRISP/Cas9 to knock in ara and arb, which showed that they could knock out genes with CRISP/Cas9 very efficiently. In this case, why did they not knock out ara and arb with CRISP/Cas9 to check consistent with whether the mutants produced by CRISP/Cas9 had the same phenotype as the mutants produced by TILLING.

Response: We appreciate the reviewer's comment. When we started this long course of study, the only available KO method for medaka was the TILLING KO. With subsequent technological development, TALEN and CRISPR/Cas9 KO came to be used. To check whether the phenotypes of the TILLING KO fish are consistent with those of other KO fish, we generated ar TALEN KO medaka lines carrying mutations leading the Ar protein truncation at its N-terminus. The resulting fish showed morphological phenotypes indistinguishable from those of the TILLING KO fish in their fin and testis. We added these data as supplementary Figures 12 and 13. Such

phenotypic consistency between the TILLING KO and TALEN KO experiments suggests that the TILLING KO phenotypes analyzed in this study are not those specifically associated with this KO method, and predicts the consistent phenotypes even in the Crispr KO experiments.

Therefore, we added the following sentence In the Discussion.

Main text, page 13, line 431-434, We confirmed that at least the phenotypes in fin morphogenesis and spermatogenesis in the *ar* TILLING KO medaka lines were reproduced by *ar* TALEN KO medaka lines (Supplementary Figs. 12 and 13), demonstrating that the observed phenotypes were not specific to the *ar* TILLING KO lines.

We also added the following sentences to the Result section to describe that the TILLING mutations in *ara* and *arb* resulted in lack of the ligand binding domain (LBD) containing the key amino acid-substitution which generates a distinct transactivation response between Ara and Arb *in vitro*.

Main text, page 5, line 173- page 6, line 175, These nonsense mutations lead to expression of the truncated protein products lacking the LBD that contains a key amino acid-substitution responsible for distinct transactivation responses of Ara and Arb *in vitro*³⁰ (Supplementary Fig. 2).

We also included the following sentences in the Supplementary Information.

Supplementary Information, page 2, line 19-40 (Supplementary Materials and Methods)

Targeted gene disruption with TALENs

Genome editing mediated by transcription activator-like effector nucleases (TALENs) was performed as previously reported². Ar proteins contain a N-terminal domain (NTD), DNA binding domain (DBD), and ligand binding domain (LBD) from the amino to carboxyl terminus. The TALEN pairs for *ara* and *arb* were designed to cleave the 1st exon of *ara* and *arb* at the downstream of their ATG start codons, respectively (their binding sequences are shown in Supplementary Fig. 12). Synthesis and purification of capped RNAs from the linealised TALEN expression vectors was performed using the mMessage mMachine SP6 kit (Thermo Fisher Scientific, Waltham, MA, USA) and the RNeasy Mini kit (Qiagen, Hilden, Germany). The pairs of RNA for TALENs were microinjected into fertilized medaka eggs at the 1-cell stage. F₀ founders were crossed with WT fish and then germline-transmitted mutant Fish (F₁) were selected by direct sequencing of the PCR products amplified from the fin clips of the adult fish using the primers, ara-F and ara-R for *ara*, and arb-F and arb-R for *arb* (Supplementary Table 1). The genetic sex of each fish was examined by genomic PCR experiments according to a previous report³. The mutant strains were

maintained by crossing heterozygous females and males, and the resulting WT and homozygous siblings were used for phenotypic analyses. The *ara* and *arb* double heterozygous males and females (*ara*^{+/-}; *arb*^{+/-}) were obtained by breeding *ara*^{-/-} males with *arb*^{-/-} females. The *ara* and *arb* double heterozygous males and females were crossed to obtain males (*ara*^{-/-}; *arb*^{+/-}) and females (*ara*^{-/-}; *arb*^{+/-}). The *ara* and *arb* double homozygous males and females (*ara*^{-/-}; *arb*^{-/-}) were obtained by breeding males (*ara*^{-/-}; *arb*^{+/-}) with females (*ara*^{-/-}; *arb*^{+/-}).

Supplementary Information, page 15, line 236-246 (Legend for Supplementary Fig. 12)

Supplementary Fig. 12

Design of transcription activator-like effector nucleases (TALENs) and frame-shift mutations in the *ara* and *arb* genes

a, c Positions of indels generated in *ar* TALEN KO fish. First ATG sequences in the 1st exon of *ara* or *arb* are in bold characters. Arrows indicate target sequences of the TALEN used in this study. Deleted nucleotides in the *ara* KO and *arb* KO fish are boxed. **b, d** Alignments of WT and mutant DNA sequences with their encoding amino acid sequences. The indels introduced by the TALENs are expected to cause frameshifts and generate premature stop codons in the Ara and Arb NTDs. Numbers on the graphical illustration of the functional domains of each Ar protein indicate the start and end residual positions of the DBD and LBD.

Supplementary Information, page 16, line 251-258 (Legend for Supplementary Fig. 13)

Supplementary Fig. 13

Fin and testicular phenotypes in *ar* TALEN KO medaka

a Representative picture of the whole body and higher magnification pictures of the anal fin and dorsal fin of a WT male, *ara* KO male, *arb* KO male, and *ar* DKO male. The papillary processes developments were marked by red dotted circles. Arrows indicate forks in the dorsal fin.

b Representative micrographs of Masson/trichrome-stained sections of adult gonads of WT males and *ar* DKO males.

We added the following sequences of PCR primers used for genotyping TALEN-targeted alleles to Supplementary Table 1.

ara-F, TGTTTGTCTCTGCGCACTC

ara-R, TGGGATCCTTGCAGATGAAT

arb-F, GCAGCAGAACTGCTCACAG

arb-R, GACACCTGTACTCGGCCACT

We added the following paper related to our TALEN experiments in References for Supplementary Information.

Supplementary Information, Page 19, line 287-290,

2. Ansai S, *et al.* Efficient targeted mutagenesis in medaka using custom-designed transcription activator-like effector nucleases. *Genetics* **193**, 739-749 (2013).
3. Matsuda M, *et al.* DMY gene induces male development in genetically female (XX) medaka fish. *Proc Natl Acad Sci U S A* **104**, 3865-3870 (2007).

2. Both *ara* and *arab* mutants are fertile, while the *ara* and *arab* double mutants can not breed with females under natural mating. The authors should examine whether a genetic compensating response has occurred.

Response: We appreciate the reviewer's comment. As the reviewer indicated, no *ar* DKO males successfully bred with females under natural mating, while the medaka lacking the single *ar* ortholog were fertile. These results leave the possibility that a genetic compensation by one of *ar* orthologs occurs in the brain responsible for the courtship behaviour. To address this question, we examined the expression of *ara* and *arb* in the whole-brain transcriptomes of *ar* KO medaka. We could not observe any significant increase of *ara* expression in *arb* KO, and *arb* expression in *ara* KO, indicating that there was no detectable level of the genetic compensation by one of *ar* orthologs at the whole-brain level.

However, this transcriptome analysis cannot exclude the possibility that the genetic compensation occurs only in a small number of brain neurons specifically involved in the courtship behaviour, though such neurons have not been identified yet in medaka. Therefore, we added the following sentences in the Results to explain the limitation of this study.

Main text, page 10, line 335 to 337, We detected no up-regulation of *ara* in the *arb* KO and that of *arb* in the *ara* KO, which indicates that no detectable level of genetic compensation occurred between *ara* and *arb* at the whole-brain level.

3. In Fig.1, the pictures with only HE staining are not enough, in situ hybridization or immune-histochemistry of some gonad specific genes must be provided, such as germ

cells gene *vasa*, Sertoli cells genes *dmy*, *gsdf* and Leydig cells gene to check why the *ara* and *arb* mutants reduce the frequency of spawning.

Response: We are grateful for this comment. Following this comment, we performed section *in situ* hybridization analysis of the gonad specific genes, *vasa* for germ cells, *P450c17* for Leydig cells, and *gsdf* for Sertoli cells in *ara* KO, *arb* KO and *ar* DKO testes. This analysis detected expression of these cell-marker genes in all of their testes, indicating the differentiation of these cell types in all *ar* KO medaka lines. We added these results as supplementary Fig. 4 and additionally wrote the following sentences in the corresponding part of the main text.

Main text, page 7, line 212-214 (Result section “Roles of the two *Ars* in testicular development and spermatogenesis”) Expression analysis of gonad-specific genes, *vasa*, *P450c17* and *gsdf*, indicates that germ cells, Leydig cells, and Sertoli cells were formed in the testes of *ar* DKO as in *ar* single KOs (Supplementary Fig.4).

We added the following sentences in Supplementary Information.

Supplementary Information, Page 2, line 13-17 (Supplementary Materials and Methods)

Section *in situ* hybridization

Section *in situ* hybridization analysis was performed as described previously¹. The cDNAs used for preparation of antisense riboprobes of *vasa*, *P450c17*, and *gsdf* were cloned by standard reverse transcription polymerase chain reaction (RT-PCR) procedures using primers shown in Supplementary Table 1.

Supplementary Information, Page 7, line 108-117 (Legend for Supplementary Fig.4),

Supplementary Fig. 4

***In situ* hybridization for gene expression analysis of testis in WT and *ar* KO strains**

Expression of *vasa* (germ cell marker) (a), *P450c17* (Leydig cell marker) (b), and *gsdf* (Sertoli cell marker) (c) in the testes of WT, *ara* KO, *arb* KO and *ar* DKO males. Hybridization reactions were performed on the cross sections of testes (8 μ m thickness). A control section incubated with a sense riboprobe was compared with a consecutive section hybridized with the antisense probe. Arrows, closed arrowheads, and open arrowheads indicate the representative expression signals of *vasa*, *P450c17*, and *gsdf*, respectively. Asterisks indicate background staining signals observed with both antisense and sense probes for the same gene. Scale bars were shown in pictures.

We added sequences of the following primers used for cloning of *gsdf*, *vasa*, and P450c17 genes to Supplementary Table 1.

gsdf-S1, TCCACCATGTCTTTGGCAC

gsdf-R1, TGACCAACCCCTGCCTAC

vasa-S1, ACGGCCCAAAGTGACCTAC

vasa-R1, GGGTCGTAGAAGGACACGG

P450c17-S1, CTCTGTGCTCCACCCTGT

P450c17-R1, GGTCTGGGTGTGGCTTTC

We added the following paper related to our section *in situ* hybridization experiments in References for Supplementary Information

Supplementary Information, Page 19, line 275-277,

1. Miyagawa S, *et al.* Dosage-dependent hedgehog signals integrated with Wnt/beta-catenin signaling regulate external genitalia formation as an appendicular program. *Development* **136**, 3969-3978 (2009).

4. Many pictures are not clear enough, such as Fig 1d, 1e and 1K, the authors should provide pictures with high resolutions.

Response: We appreciate the reviewer's comment. We replaced the figures with those with higher resolutions in the revised manuscript set.

5. The ms should have more words about the knock in plasmids, such as what is mClover3, why the fish was in green?

Response: We appreciate the reviewer's comment.

We added the explanation of mClover3 and P2A sequence in the Materials and Methods "Generation of Ar-FLAG-mClover3 knock-in (KI) medaka strains", as follows.

Main text, page 18, line 629-630, mClover3 (a brighter and more photostable variant of green fluorescent protein)⁸¹

Main text, page 18, line 639-page 19, line 644, Each donor vector was inserted into a gRNA-targeting site in the 8th or 7th intron of *ara* or *arb*, respectively. The resulting alleles could

express mRNAs under the control of endogenous *ar* promoters, and the mRNAs are expected to be translated into two distinct proteins, namely, the C-terminal FLAG-tagged Ar and mClover3, owing to a ribosomal skipping by the P2A sequence⁸².

We edited the origin of 2A peptide in following part in the legend for Figure 6. We used P2A (2A peptide from porcine teschovirus-1), but not T2A (2A peptide from thosea asigna virus 2A).

Main text, page 33, line 1157-1158, a P2A (2A peptide from porcine teschovirus-1)-mClover3

Main text, page 34, line 1159-1160, Both AR-FLAG and P2A-mClover3

We added the following papers in the References.

Main text, page 28, line 998- page 29, line 1003

81. Bajar BT, *et al.* Improving brightness and photostability of green and red fluorescent proteins for live cell imaging and FRET reporting. *Sci Rep* **6**, 20889 (2016).
82. Inoue T, Iida A, Maegawa S, Sehara-Fujisawa A, Kinoshita M. Generation of a transgenic medaka (*Oryzias latipes*) strain for visualization of nuclear dynamics in early developmental stages. *Dev Growth Differ* **58**, 679-687 (2016).

6. Fig legends of 1e are not in consistent with Fig.1e

Response: We appreciate the reviewer's comment. This was our mistake.

We have amended the term "*ara*^{-/-}; *arb*^{-/-}" to "*ar* DKO" in the legend of Figure 1e.

7. In the ms, "fertilisation" should be "fertilization"; "localization" should be "localization"; "Visualisation" should be "Visualization".

Response: We thank this careful comment. We used the British English.

8. Line 540, six month medaka are too old, they are not good for the experiment, since 3 month old medaka are matured.

Response: We appreciate the reviewer's comment.

Based on the life span analyses of medaka by long cultured data (Gopalakrishnan *et al.* 2013), we judged that the 6 months old is a stably reproducing age. Gopalakrishnan *et al.* (2013) reported that the plasma T and 11-KT levels peaked in males at 8 months old and gradually decreased with age after 8 months. They also

reported that the median life span of the males and females under laboratory condition was 13.7 months and 14.6 months, respectively. In consistent with these reports, in our laboratory condition, males constantly reproduce at 6 months old. In addition, as we described in the Materials and Methods, we used the male and female medaka producing fertilised eggs for at least three consecutive days until the day before the experiment were used. The *ar* DKO males were also reared with females in the same tank for one week until the day before the experiment.

We added the information of the age of medaka into the Materials and Methods as follows,

Main text, page 17, line 596-598, We judged that the 6 months old was proper to check the testicular function of medaka, because it has been shown that the plasma T and 11-KT levels peak in male medaka at 8 months old⁷⁹.

We added the following paper in the References.

Main text, page 28, line 992-995

79. Gopalakrishnan S, Cheung NK, Yip BW, Au DW. Medaka fish exhibits longevity gender gap, a natural drop in estrogen and telomere shortening during aging: a unique model for studying sex-dependent longevity. *Front Zool* **10**, 78 (2013).

9. Full name of 'WT' should be given as it appears for the first time.

Thank you for this comment. We added the full name before the abbreviation when it appeared for the first time in the main text line 187 (page 6), we showed WT as follows, "All wild-type (WT) and mutant fish with a Y chromosome (XY chromosomes) had testes (n = 10 in each genotype)".

Reviewer #2 (Remarks to the Author):

The current manuscript analysed in detail the unique and overlapping functions of two androgen receptor (*ar*) orthologs in various sexual characteristics in one of the teleost model species, the Japanese medaka. The authors used both mutants for either *ar* copies (*ara* KO and *arb* KO) and double mutants (*ar* DKO) to compare the effect resulting from the loss of either *ar* gene. A wide range of sexual characteristics were examined carefully, including mutant fecundity, morphology, mating success, behaviour and brain gene expression patterns. With these comparisons, the authors identified distinct

functions of either ar or ohnolog. Furthermore, the authors used epitope-tagged AR-KI mutants show that the functional difference between the two ohnologs potentially arise from their different transcriptional regulation and intracellular localisation. Overall, this manuscript is very logically structured and easy to read with clear writing. I find the experiments conducted and analyses performed thorough and convincing. I especially enjoyed the included video of aggressive competition among the male fish. I do not have major criticism on this manuscript, but have a few comments that I hope the authors can address to further improve the manuscript.

I couldn't find information regarding the levels of divergence between ara and arb. Do they share functional domains ?

Response: We are grateful for this comment. Medaka Ara and Arb share the three major functional domains, a hypervariable N-terminal domain (NTD), a highly conserved central DNA binding domain (DBD) consisting of two zinc finger motifs, and a COOH-terminal ligand binding domain (LBD). Comparison of the deduced amino acid sequences of the DBD and LBD revealed that medaka Arb has higher similarity to human AR compared with that of medaka Ara (We added these data in Supplementary Fig. 1a.). The higher divergence in teleost Ara sequences, as judged by a long branch in the molecular phylogenetic tree (We added the tree data modified from Ogino et al. 2016³⁰ by taking the permission to reuse in Supplementary Fig. 1), indicates that, after the duplication that gave rise to Ara and Arb, the coding sequence of Ara accumulated novel substitutions at a greater rate than that of Arb. In fact, we previously revealed that Ara acquired a new function as a hyperactive form of Ar, showing higher ligand-dependent transactivation capacity and constitutive nuclear localisation activity *in vitro*²⁵. We also found two key nonsynonymous base substitutions that have been conserved in the hinge region and LBD of spiny-rayed fish (Acanthomorpha) Aras including medaka Ara³⁰. Such substitution in the hinge region has changed the molecular property of Ara from a ligand-dependent- to a constitutive nuclear localisation protein, while the substitution in the LBD has increased its transactivation capacity *in vitro*³⁰.

Following the reviewer's comment, we have included the graphical illustration of the functional domains of the Ars, the similarity of deduced amino acid sequences of each domain of medaka Ara and Arb to those of human AR, and molecular phylogenetic tree of vertebrate Ar genes in Supplementary Fig 1. We additionally

wrote the following sentences in the corresponding part of the main text and the Supplementary Information.

Main text, page 3, line 87-90 (Introduction)

The Ar is composed of three major functional domains, a hypervariable N-terminal domain (NTD), a central highly conserved DNA binding domain (DBD) consisting of two zinc finger motifs, and a COOH-terminal ligand binding domain (LBD)¹⁷⁻²⁰ (Supplementary Fig. 1).

Main text, page 4, line 120 (Introduction)

Since the abbreviation was shown earlier in line 90, “ligand-binding domain (LBD)” in line 121 was changed to “LBD”.

Main text, page 5, line 171-page 6, line 175 (Result section “Screening of *ara* and *arb* knockout mutants (KOs)”)

Screening of *ara* and *arb* knockout mutants (KOs)

By screening the medaka TILLING library, we identified founders possessing nonsense mutations in exon 6 of *ara* (S507X) and exon 4 of *arb* (L503X) (Supplementary Fig. 2a–c). These nonsense mutations lead to expression of the truncated protein products lacking the LBD that contains a key amino acid-substitution responsible for distinct transactivation responses of Ara and Arb *in vitro*³⁰ (Supplementary Fig. 2).

Supplementary Information, page 3, line 44-59 (Legend for Supplementary Fig.1),

Supplementary Fig. 1

Phylogenetic analysis of Ar genes

a Structures of medaka Ara (GenBank accession number: AB252233) and Arb (AB252679) proteins. Ar is composed of three major functional domains, a hypervariable N-terminal domain (NTD), a central highly conserved DNA binding domain (DBD) containing two zinc finger motifs, and a COOH-terminal ligand binding domain (LBD). The numbers above each box refer to the position of amino acids in the putative DBD and LBD. The % of identity of deduced amino acid sequences of each domain to human AR (NM_000044.2) is shown in boxes.

b A molecular phylogenetic tree of vertebrate *ar* genes. This tree was estimated with AR protein sequences using the maximum-likelihood (ML) method combined with JTT substitution model. The Brown-banded bambooshark *ar* gene was used as an outgroup. Support values at nodes are bootstrap probabilities in the ML analysis.

c GenBank and Ensembl accession numbers of the gene sequences used in this analysis and their species names. This figure was modified from Ogino et al. 2016⁴ by taking a permission to reuse

(license Number 5411120388057).

Supplementary Information, page 4, line 62-69 (Legend for Supplementary Fig.2),

Supplementary Fig. 2

Identification and verification of *ara/arb* TILLING mutant medaka

a Protein structures of medaka Ara and Arb, and positions of the mutations identified in this study. An amino acid in the Arb LBD, F702, is conserved in ancestral-type tetrapod Ar proteins. The corresponding amino acid is replaced with Y634 in Ara, which is responsible for different ligand-dependent transcriptional responses of Ara and Arb *in vitro*⁴. The nonsense mutations in *ara* (S507X) and *arb* (L503X) resulted in expression of the truncated protein products lacking the LBD.

Are they located in syntenic blocks in the genome?

Response: We are grateful for this comment. To judge whether *ara* and *arb* were generated by a whole genome duplication event or a chromosomal tandem duplication event, we have previously compared their genome synteny in medaka with that of human *AR*. The medaka *ara* and *arb* are located on chromosomes 10 and 14, respectively. The human *AR* is located on chromosome X. Flanking regions of *ar* genes on medaka chromosomes 10 and 14 contain genes orthologous to those located in syntenic regions on human chromosome. Therefore, we concluded that the teleost *ar* gene duplication occurred in association with the teleost-specific whole genome duplication. We published these data in 2009 (Ogino et al. *Endocrinology* 150, 5415-5427, 2009). As suggested by the reviewer, we have now included this information in the Introduction as follows.

Main text, page 3, line 99-102, The medaka *ara* and *arb* were mapped to chromosomes 10 and 14, respectively, with a conserved synteny relative to a single region locating the *AR* gene in human chromosome X, suggesting that the teleost *ar* gene duplication occurred as the result of TSGD²⁵.

Do they have very divergent putative promoter sequences that might be underlying the different transcriptional regulation?

We are grateful for this comment.

We aligned a 20-kb medaka genomic sequence corresponding to *ara* and *arb* genes and the 5' upstream and 3' downstream regions with their orthologous sequences

from stickleback, fugu, zebrafish, and non-coding ATAC-seq peak sequences associated with the medaka *ara* and *arb* using MultiPipMaker. The ATAC-seq data were obtained from http://tulab.genetics.ac.cn/medaka_omics/ (Li, Y. et al., 2020). The resulting percent identity plot (pip) views indicate that part of the ATAC-seq peak regions associated with medaka *ara* is conserved in the stickleback and/or fugu *ara* genome sequence(s), but not in the zebrafish *ar* or medaka *arb* genome sequences. A single ATAC-seq peak region associated with medaka *arb* is conserved in the stickleback and fugu *arb* genome sequences, but not in the zebrafish *ar* or medaka *ara* genome sequences. These alignments suggest that *cis*-regulatory mechanisms of *ara* and *arb*, respectively, are partially conserved in the teleost species that retain both *ar* ohnologs, but not in the species that lack one or the other, and are not conserved between *ara* and *arb*. Consistent with this sequence alignment analysis, Ara and Arb expression patterns visualized by green fluorescence in Ar knock-in (Ar-KI) males were different. The Ara-KI males showed weak fluorescence throughout the trunk and fins, and strong fluorescence in the regions adjacent to the pectoral, dorsal and anal fins. In contrast, the Arb-KI males showed a more restricted pattern of green fluorescence that localized primarily to the pectoral, dorsal and anal fins. We have now included the results of comparison of *cis*-regulatory sequences in Supplementary Fig 11 and wrote the following sentences in the Results.

Main text, page 10, line 345-349 (Result section “Brain transcriptomic changes in *ar* KOs”)

Comparative genome sequence analysis combined with the available ATAC-seq data in Medaka⁵² shows that putative *cis*-regulatory sequences of *ara* and *arb*, respectively, were partially conserved in teleost species, but not between *ara* and *arb*, supporting their subfunctionalization and/or neofunctionalization (Supplementary Fig. 11).

Main text, page 11, line 356-363 (Result section of “Differences in expression and intracellular localisation of the two Ars”)

The Ara-KI adult males showed weak green fluorescence throughout the trunk and fins, and strong fluorescence in the regions adjacent to the pectoral, dorsal and anal fins. In contrast, the Arb-KI adult males showed a more restricted pattern of green fluorescence that localized primarily to the pectoral, dorsal and anal fins. (Fig. 6b). In the papillary processes of the anal fin, stronger green fluorescence was observed in Arb-KI than in Ara-KI males (Fig. 6b). These differences in the expression patterns of Ara and Arb appear to be consistent with the results obtained from the comparative analysis of their *cis*-regulatory sequences.

Main text, page 34, line 1169-1172 (Legend for Figure 6b)

White, gray, and red arrows indicate the regions adjacent to pectoral, dorsal, and anal fins, respectively (Ara-KI). White, gray, and red arrows indicate the pectoral, dorsal, and anal fins, respectively (Arb-KI).

Supplementary Information, page 13, line 200-page 14, line 232 (Legend for Supplementary Fig.11),

Supplementary Fig. 11

Comparison of *cis*-regulatory sequences of *ar* genes in teleost

a, A 20-kb medaka genomic sequence corresponding to *ara* gene and the 5' upstream and 3' downstream regions (oryLat2, chr10: 18,344,871 – 18,364,872) was aligned with its orthologous sequences from stickleback (gasAcu1, chrIV: 15,180,906 – 15,200,410), fugu (fr3, chr14: 7,895,689 – 7,913,481), zebrafish (danRer10, chr5: 34,924,025 – 35,123,662), a medaka genomic sequence corresponding to *arb* gene and the 5' upstream and 3' downstream flanking regions (ASM223467v1, chr14: 16,689,824 – 16,772,825), and non-coding ATAC-seq peak sequences associated with the medaka *ara* gene (Li, Y. et al., 2020²; http://tulab.genetics.ac.cn/medaka_omics/) using MultiPipMaker to generate a percent identity plot (pip) view³. A blue arrow indicates *ara* with its transcriptional orientation. Green and yellow shadings indicate *ara* exons and introns, respectively. Part of the ATAC-seq peak regions is conserved in the *ara* genome sequences of medaka, stickleback, and/or fugu (shaded in magenta) but not in the zebrafish *ar* or medaka *arb* genome sequences. **b**, An approximately 20-kb genomic sequence corresponding to the first, second, and third exons of *arb* gene and the 5' upstream region (oryLat2, chr14:17,246,771 – 17,266,732) was aligned with its orthologous sequences from stickleback (gasAcu1, chrVII:17,122,259 – 17,143,909), fugu (fTakRub1.2, chr15:12,602,150 – 12,632,151), zebrafish (danRer10, chr5: 34,924,025 – 35,123,662), a medaka genomic sequence corresponding to *ara* gene and the 5' upstream region (ASM223467v1, chr10: 22,174,238 – 22,194,238), and non-coding ATAC-seq peak sequences associated with the medaka *arb* gene (Li, Y. et al., 2020²; http://tulab.genetics.ac.cn/medaka_omics/) using MultiPipMaker. Note that the ATAC-seq peaks associated with the medaka *arb* gene localize only within this 20-kb region. A blue arrow indicates *arb* with its transcriptional orientation. Green and yellow shadings indicate *arb* exons and introns, respectively. Only one peak region of ATAC-seq is conserved in the *arb* genome sequences of medaka, stickleback, and fugu (shaded in magenta) but not in the zebrafish *ar* or medaka *ara* genome sequences. These alignments suggest that *cis*-regulatory mechanisms of *ara* and *arb*, respectively, are partially conserved in the teleost species that retain both *ar* ohnologs, but not in the species that lack one or the other, and are not conserved between *ara* and *arb*. The genome sequences used here were downloaded from the UCSC Genome

Browser and Ensembl Genome Browser^{4, 5}.

**We added the following paper in the References for Supplementary Information
Supplementary Information, page 19, line 284-291**

5. Li Y, Liu Y, Yang H, Zhang T, Naruse K, Tu Q. Dynamic transcriptional and chromatin accessibility landscape of medaka embryogenesis. *Genome Res* **30**, 924-937 (2020).
6. Schwartz S, *et al.* MultiPipMaker and supporting tools: Alignments and analysis of multiple genomic DNA sequences. *Nucleic Acids Res* **31**, 3518-3524 (2003).
7. Kent WJ, *et al.* The human genome browser at UCSC. *Genome Res* **12**, 996-1006 (2002).
8. Martin FJ, *et al.* Ensembl 2023. *Nucleic Acids Res* (2022).

The authors included the discussion of identification of the downstream effector genes of *Ars* (Line420-line434), but I didn't find the relevant results for this investigation in the result section "Brain transcriptomic change in *ar* KOs". Maybe consider restructuring the presentation of these results and discussion.

Response: We appreciate the reviewer's comment.

We added the following sentences in the Result section of "Brain transcriptomic changes in *ar* KOs" .

Main text, page 10, line 329-335, Among these genes, expression level of neuropeptide B a (*npba*), which is known to regulate the female reproductive behaviour⁴⁹, was 3.98 and 5.17 times higher in the *arb* KO and *ar* DKO males compared to WT males, respectively (Fig. 5g, i). Additionally, expression level of *hsd17b12a*, whose product catalyses the transformation of estrone (E1) into E2⁵⁰ and 11-ketoandrostenedione (11KA4) to 11KT⁵¹, was significantly higher in the males of all *ar* KO strains compared to WT males (Fig. 5g-i).

Main text, page 10, line 338, We changed "oestron (E1)" to "E1", because the full name of abbreviation appeared in added sentences in line 333.

Main text, page 10, line 340-345, These results indicate that *arb* predominantly represses the *npba* expression, whereas both *ar* ohnologs are required for repressing *hsd17b12a* expression in males. This is likely the reason why 11KT level increased in *ar* DKO males. Such distinct roles for *ar* ohnologs in the regulation of *Ar*-biased gene expression in the brain may reflect the subfunctionalisation and/or neofunctionalisation of these ohnologs after the genome duplication

event.

We added the following paper in References.

Main text, page 26, line 921-924,

51. Suzuki H, Ozaki Y, Ijiri S, Gen K, Kazeto Y. 17beta-Hydroxysteroid dehydrogenase type 12a responsible for testicular 11-ketotestosterone synthesis in the Japanese eel, *Anguilla japonica*. *J Steroid Biochem Mol Biol* **198**, 105550 (2020).

More on this section, line295-lin296 , the authors wrote ‘ suggesting the occurrence of a small number of genes exhibiting low expression but having large effect on neuronal signal’. I find the support to this strong conclusion rather weakly supported, and it is not clear if it is based on the GO enrichment analysis or the fact that there are 290 DE genes? Is 290 genes fewer than expected compared to other studies analysing gene expression change related to behaviour? Could it be that the brain gene expression would be more or less exaggerated based on the context? Maybe consider rephrasing?

Response: We appreciate the reviewer’s comment. We agree with the reviewer’s suggestion. We could not detect the particular biological processes by Gene Ontology (GO) enrichment analysis, although we identified 290 genes that were differentially expressed between the WT and *ar* DKO males, indicating that whole brain transcriptome analysis provides insufficient resolution to identify the genes that regulates the courtship behaviour. Higher resolution analysis at the cellular level, an identification of the neurons that regulates the male reproductive behaviour and gene expression analysis in such specific neurons would be necessary to identify the genes that regulate male courtship display. Therefore, we have rephrased the sentences to explain the limitation of this study as follows.

Main text, page 9, line 318 - page 10, line 320 (Result section “Brain transcriptomic changes in *ar* KOs”)

Identification of genes regulating the male courtship display would require identification of neurons that control the male reproductive behaviour and high-resolution gene expression analysis in such neurons.

Line190 and Line 191 present contradictory ideas, potential a typo?

Response: We appreciate the reviewer's comment. We have amended the sentence of the corresponding part as follows.

Main text, page 6, line 208-209, These results indicate that all the mutant strains produce functional sperms that were capable of fertilising the eggs.

Line 354- line 358: I'm not quite following how ar signalling resolves sexual conflict in medaka, could the author elaborate on this idea? 'Acquisition of exaggerated male-specific traits in the lineages leading to medaka' would be more convincing if the author comment on the level of sexual dimorphism in the lineages leading to medaka first.

Response: We greatly appreciate the reviewer's comment.

Because Ar signaling is dispensable for female fecundity but is indispensable for male fecundity by regulating the proper male specific reproductive behaviour and external morphology, we tried to suggest that androgen-dependent regulation would be able to solve the sexual conflict by regulating the male-biased phenotypes. However recent analysis indicates that sex linkage which allows males and females to carry different alleles on sex chromosomes is more effective than androgen regulation in the production of large sex differences in gene expression, and androgen-dependent regulation can contribute to temporary resolution of sexual conflict in stickleback¹⁴, Therefore, we cannot simply suggest that the androgen/Ar system resolves sexual conflict in medaka based on ar KO phenotypes. Therefore, we deleted "suggesting that the androgen/Ar system resolves sexual conflict in this species." from this sentence.

Main text page 12, line 399-400, In contrast, Ar signalling is dispensable for female fecundity in medaka, ~~suggesting that the androgen/Ar system resolves sexual conflict in this species.~~

We also added the explanation about the acquisition of exaggerated male-specific traits in the lineages leading to medaka as follows (Main text, page 17, line 403-405 in the Discussion)

Main text, page 12, line 399-405, In contrast, Ar signalling is dispensable for female fecundity in medaka. The loss of contribution of Ar signalling in gametogenesis in both sexes might have reduced the evolutionary constraints on the ar orthologs and accelerated the acquisition of exaggerated male-specific traits in the lineage leading to medaka. In fact, the medaka species that are widely

distributed in throughout southern and southeast Asia exhibit diversification of male-specific sexual characteristics in their fin morphology, pigmentation and behaviours^{35, 38, 39, 41, 61, 62}.

Line 450-452: “resolved the pleiotropic function derived from their ancestral gene”. I couldn’t find relevant information in the manuscript for the function of *ar* in species that branched prior to TSGD to support this statement on ‘ancestral’ state.

Response: We greatly appreciate the reviewer’s comment.

To definitively determine the function of ancestral *ar* gene before the TSGD, further studies of earlier branching teleost with two *ar* homologs such as Japanese eel and basal non-teleost ray-finned fishes such as *Polypterus* possessing a single *ar* gene are required (as we mentioned in **main text, page 13, line 452-455**), however it is difficult to knockout the *ar* gene in such non-model species and entire coding sequence of *Ar* has not been identified yet in *Polypterus*. Therefore, we focused on the protein property of *Ars* in Elopomorpha lineage including the Japanese eel that represents the earlier branching teleost groups⁶⁷. Our previous analysis indicated that the Japanese eel have *ar* genes derived from ancestral genes before functional diversification, because the Japanese eel *Ara* and *Arb* did not show significant difference in their protein property. In addition, the insertion of the key substitution generating the new functionality of spiny-rayed fishes *Ara* into *Ars* from Japanese eel recapitulates the evolutionary novelty of spiny-rayed fish *Ara in vitro*³⁰. The gene expression analysis of Japanese eel *ars* indicates that they have pleiotropic expression in various tissues such as testis, muscle, and spleen⁶⁷. These observations suggest that the functional diversification of *ar* ohnologs has occurred in teleosts after the divergence of Elopomorpha lineage.

In our manuscript, we added the explanation about the protein property of Japanese eel *Ars* in the introduction and the phylogenetic tree of *Ars* in Supplementary Fig 1.

Main text, page 4, line 111-115 (Introduction)

Molecular evolutionary analysis of the teleost lineage has revealed the asymmetric evolution of *ar* ohnologs, including the accumulation of more novel substitutions in *ara* than in *arb* after the divergence of Elopomorpha, and that the lineage-specific loss of *ara* occurred independently in Otocephala such as zebrafish and Salmoniformes such as rainbow trout (Supplementary Fig. 1)^{24, 25, 29-31}.

Main text, page 4, line 119-121 (Introduction)

We also found two key nonsynonymous base substitutions in the Ar hinge region and LBD, which are highly conserved among spiny-rayed fish (Acanthomorpha) Aras, including medaka and cichlid Aras **but not in Japanese eel Ara**³⁰.

We also additionally wrote following sentences in the Discussion.

Main text, page 13, line 455-462,

So far, our previous analysis indicates that the Elopomorpha lineage including Japanese eel that represents the earlier branching teleost groups^{59, 68} have *ar* genes derived from ancestral genes before functional diversification, because the Japanese eel Ara and Arb did not show significant difference in their protein property *in vitro*³⁰ and have pleiotropic expression in various tissues such as testis, muscle, and spleen⁶⁹. These observations suggest that the functional diversification of *ar* ohnologs has occurred in teleosts after the divergence of Elopomorpha lineage.

Reviewer #3 (Remarks to the Author):

Overall feedback:

This manuscript describes a very thorough study in which the authors sought to identify the molecular properties of androgen receptor ohnologs underpinning the diversification of sexual characteristics in a fish, the Japanese medaka (*Oryzias latipes*). As such, this paper provides important insights into the mechanisms of radiation of teleost fishes driven by sexual selection.

I enjoyed reading this article. It covers an important topic using a study species that is less conventional than, for example, rodents, which have been used extensively in studies of sexual dimorphism. I also appreciate that the authors tested for effects of deficiency of androgen receptors alpha and beta on a broad range of morphological, reproductive, and behavioural traits (e.g. fin size and shape, tooth enlargement, spermatogenesis, courtship behaviour).

Because of the high standard and thoroughness of the study and manuscript, I have quite few comments and recommend a minor revision. However, I do think that these points will help to improve the manuscript.

Line-by-line feedback:

Line 57: I am not sure about the use of the phrase ‘behavioural attractiveness’ here given that ‘attractiveness’ is perceived by the behavioural signal receiver, which depends on a whole host of other factors. Is there a more accurate term that could be substituted in here

Response: We agree with the reviewer’s comment. We have switched the phrase to “the reproductive behaviour eliciting female receptivity” in the revised manuscript. We have amended the corresponding sentence as follows.

Main text, page 2, line 56-58 (Abstract)

ara was required for tooth enlargement and the reproductive behaviour eliciting female receptivity, while *arb* for male-specific fin morphogenesis and sexual motivation.

Main text, page 13, line 427-431 (Discussion)

Taken together, we revealed the differential roles of *ar* ohnologs associated with the unique sexual characteristics of teleosts—*ara* predominantly regulates the masculinisation of teeth and the reproductive behaviour eliciting female receptivity while *arb* plays essential roles in male-specific fin morphogenesis and sexual motivation.

Line 62: No capital is needed for ‘first’.

Response: We thank this careful comment. As the reviewer suggested, we have amended “First” to “first”.

Lines 131–132: It would be useful here to just briefly describe the types of actions that are typical of male medaka reproductive behaviour.

Response: We are grateful for this comment. Following the reviewer’s comment, we have now described the typical mating behaviours of male and female medaka in the Introduction section as follows.

Main text, page 5, line 141-147, The sequence begins with the male approaching and following the female closely. The male then performs a courtship display, in which he swims quickly in a

circular pattern in front of the female. If the female accepts the male, the male grasps her with his fins (termed “wrapping”), and they quiver together until eggs and sperm are released (“spawning”). If the male is not accepted, she either rapidly moves away from the male or assumes a rejection posture^{38,41}.

Line 269–272: The term ‘wrapping’ has not yet been defined and so the reader will not have proper context for this statement. It would be useful to earlier define this term and, in addition, briefly describe the typical mating behaviour of male and female medaka.

Response: We are grateful for this comment. The behaviour that the male grasps the female with his fins is termed “wrapping”. Following this comment, we described the typical mating behaviours of both sexes in the Introduction as we have shown above.

Lines 682–683: ‘The number of males escaping wrapping by the male was counted as the number of wrapping rejected by the female.’ Again, these behaviours are not sufficiently described and so will be potentially confusing to readers.

Response: We agree with the reviewer’s comment. We have amended the corresponding part in the Materials and Methods and Figure legends as follows.

Main text, page 21, line 750–page 22, line 753 (Materials and Methods “Behavioural assays”),

Wrapping behaviours that did not lead egg spawning were counted as wrapping rejections. If the wrapping led the spawning successfully, the period from the initiation of wrapping to the end of spawning was measured as a wrapping duration.

Main text, page 33, line 1136 (legend for Figure 5)

“Duration of wrapping and spawning” has amended to **“Duration of wrapping followed by spawning”**

Line 684–685: How were the behaviours recorded? Using a behaviour-logging software? If so, the name and version of the software should be provided. Also, was the observer blind to treatment while scoring trials?

Response: We appreciate the reviewer’s comment. We recorded the behaviour using a digital video camera HDR-PJ800 (Sony, Tokyo, Japan) and then counted each

behaviour by watching the recorded movie files. To distinguish the video files, line name indicated by number was recoded in video files, but we did not provide the information of each medaka lines to observer.

Line 695: ‘The mate choice test was performed using a round-robin tournament with five males.’ It is not clear what this means. Please provide sufficient detail and clarity when describing all behavioural assays so that there is no chance of misinterpretation by readers.

Response: We appreciate the reviewer’s comment. We added the detail explanation of the behavioral analysis by changing the sentences in the Materials and Methods “Behavioural assays” as follows.

Main text, page 22, line 766-785,

To directly measure the female mate preference to WT male or *ar* KO male, we performed round-robin tournaments using five WT males and five *ar* KO males, that is, a total of 25 combinations of mate choice test for each *ar* KO strain. The scores were given as the frequency with which each male mated with a female. Similar to the 1-to-1 mating behavioural analysis above, this mate choice test was initiated after the onset of the light period (8:30–10:00). A WT male and an *ara* KO or *arb* KO male were transferred to a single tank with a WT female. Then, their behavioural interactions were recorded until spawning. Each male individual was distinguished by the pigment pattern of the melanophores and fin shape. A male individual that led to egg spawning was judged as a successful mating partner under observation of each video recording. If the female did not lay eggs within 1 hour of a behavioural trial, we judged the trial to be a draw. Effects of each *ar* mutation on the mate behaviour were statistically analysed using the chi-squared (χ^2) test of independence.

To measure the female fecundity of the *ar* KO strains, a female of WT, *ara* KO, *arb* KO, or *ar* DKO strains was transferred into a test tank with a WT male, and they were separated by a partition on the day before the test, as described above. After removing the separator in the morning of the test day, we analysed whether the females spawned the eggs. If the female successfully spawned, the total number of eggs obtained and their fertilisation rates were quantified. This test was repeated for 10 consecutive days by exchanging the males daily.

Line 713: I recommend that all data and code, including for behavioural trial data, are made publicly available alongside the article. In particular, uploading all of the data that

would be required to re-run your analysis is crucial to making your research transparent.

Response: We appreciate the reviewer's comment. We uploaded all plot data in a single Excel file (Source data_MCOMMS-22-27469).

We are also ready to upload the huge video data of the behavior tests (100GB). We would be happy to do it immediately if the editorial office could kindly tell us an appropriate server for Nature Communications.

We also revised the behavioural analysis data of *ar* DKO males for a more reliable comparison with those of WT and *ar* single KO males, because the number of their behavioural tests was smaller than those of WT and single *ar* KO males in our previous manuscript (13 tests in *ar* DKO, 52 tests in WT, 38 tests in *ara* KO, 33 tests in *arb* KO). In the revised data, we analyzed 43 mating tests for *ar* DKO males (30 minutes each), and found that *ar* DKO males took courtship behaviour in 6 mating tests, although the frequency of this behaviour was very few, once in the 4 tests and twice in the 2 tests. These results indicate that *ar* DKO males mostly, but not completely, abolished courtship displays. Therefore, we amended the corresponding texts of the revised manuscript as follows.

Main text page 8, line 273-276 (Result section "Differential effects of the two *Ars* on mating behaviours")

The *ar* DKO males **mostly** abolished courtship displays during behavioural testing, whereas the single mutant males displayed courtships **in over 90% of the tests** (Fig. 5a), indicating that either of the two *ars* is sufficient for this mating behaviour.

Main text page 11, line 382-385 (Discussion)

We demonstrated that *ar* DKO males **mostly abolished** courtship displays and **lacked** external sexual characteristics, such as masculinisation of fin morphology and pigment cell patterns, resulting in infertility.

Main text page 20, line 702-703 (Materials and Methods section "RNA isolation")

ar DKO males ~~that lack courtship~~ were anaesthetised after 5 min of mating with a WT female.

We revised the *ar* DKO data in Fig. 5a (Percentage of mating tests with different females in which a male exhibited courtship display within a 30 min).

We showed the WT data for comparison with those of *ara* KO or *arb* KO from the

same litter.

We added the number of the behavioural tests that we analysed in the Figure legends for Fig. 1a and Fig. 5a as follows. We also summarized the number of the mating tests for each analysis in supplementary table 2.

Main text, page 31, line 1055-1060 (Legend for Fig. 1a “Frequency of mating tests in which a WT female laid eggs within 30 min after mating”)

We used WT males (n = 11, total 60 mating tests) and *ara* KO males (n = 14, total 76 mating tests) of the same litter, and WT males (n = 6, total 42 mating tests) and *arb* KO males (n = 6, total 41 mating tests) of the same litter, and *ar* DKO males (n = 10, total 43 mating tests), as mating partners for the WT females. Details of mating tests utilized for this and following behavioural analyses were shown in supplementary Table 2.

Main text, page 33, line 1128-1132 (Legend for Fig. 5a “Percentage of mating tests with different females in which a male exhibited courtship display within 30 min”)

We used WT males (n = 11, total 59 mating tests) and *ara* KO males (n = 14, total 76 mating tests) of the same litter, and WT males (n = 6, total 40 mating tests) and *arb* KO males (n = 6, total 40 mating tests) of the same litter, and *ar* DKO males (n = 10, total 43 mating tests) as mating partners for the WT females.

Main text, page 22, line 753-754 (Materials and Methods section “Behavioural assays”) We summarized the number of the mating tests for each analysis in supplementary table 2.

The edited values and other rhetorical corrections are shown in red in the text.

Again, we thank reviewers’ comprehensive comments which were very useful for the improvement of our manuscript. We hope the revised version of the manuscript is acceptable to the Nature Communications.

Sincerely yours,

Yukiko Ogino

Associate Professor

Laboratory of Aquatic Molecular Developmental Biology,

Graduate School of Bioresource and Bioenvironmental Sciences,

Kyushu University
Fukuoka 819-0395, Japan

REVIEWERS' COMMENTS

Reviewer #1 (Remarks to the Author):

The revised version has addressed all my concerns and thus is acceptable for publication.

Reviewer #2 (Remarks to the Author):

I'm satisfied with how the authors addressed my comments and I'd be happy to see this manuscript published.

Reviewer #3 (Remarks to the Author):

The authors have adequately addressed my comments. I recommend acceptance of the paper, which is contingent on the other reviewers also feeling that their comments have been adequately dealt with.